



# The Adaptable 4A Inversion (5AI): Description and first $XCO_2$ retrievals from OCO-2 observations

Matthieu Dogniaux[1], Cyril Crevoisier[1], Raymond Armante[1], Virginie Capelle[1], Thibault Delahaye[1], Vincent Cassé[1], Martine De Mazière[2], Nicholas M. Deutscher[3,4], Dietrich G. Feist[5,6,7], Omaira E. Garcia[8], David W. T. Griffith[3], Frank Hase[9], Laura T. Iraci[10], Rigel Kivi[11], Isamu Morino[12], Justus Notholt[4], David F. Pollard[13], Coleen M. Roehl[14], Kei Shiomi[15], Kimberly Strong[16], Yao Té[17], Voltaire A. Velazco[3], Thorsten Warneke[4]

[1]Laboratoire de Météorologie Dynamique/IPSL, CNRS, École polytechnique, Institut Polytechnique de Paris, Sorbonne Université, École Normale Supérieure, PSL Research University, Palaiseau, 91120, France
[2]Royal Belgian Institute for Space Aeronomy, Brussels, Belgium
[3]Centre for Atmospheric Chemistry, School of Earth, Atmospheric and Life Sciences, University of Wollongong, Wollongong, Australia
[4]University of Bremen, Bremen, Germany
[5]Max Planck Institute for Biogeochemistry, Jena, Germany
[6]Ludwig-Maximilians-Universität München, Lehrstuhl für Physik der Atmosphäre, Munich, Germany
[7]Deutsches Zentrum für Luft- und Raumfahrt, Institut für Physik der Atmosphäre, Oberpfaffenhofen, Germany
[8]Izaña Atmospheric Research Center (IARC), State Meteorological Agency of Spain (AEMET), Spain
[9]Karlsruhe Institute of Technology (KIT), Institute of Meteorology and Climate Research (IMK-ASF), Karlsruhe, Germany
[10]NASA Ames Research Center, Moffett Field, CA, USA
[11]Finnish Meteorological Institute, Sodankylä, Finland
[12]National Institute for Environmental Studies (NIES), Tsukuba, Japan
[13]National Institute of Water and Atmospheric Research Ltd (NIWA), Lauder, New Zealand
[14]Division of Geological and Planetary Sciences, California Institute of Technology, Pasadena, CA, USA
[15]Japan Aerospace Exploration Agency (JAXA), Tsukuba, Japan
[16]Department of Physics, University of Toronto, Toronto, Canada
[17]Laboratoire d'Etudes du Rayonnement et de la Matière en Astrophysique et Atmosphères (LERMA-IPSL), Sorbonne Université, CNRS, Observatoire de Paris, PSL Université, 75005 Paris, France

*Correspondence to*: Matthieu Dogniaux (matthieu.dogniaux@lmd.ipsl.fr)

**Abstract.** A better understanding of greenhouse gas surface sources and sinks is required in order to address the global challenge of climate change. Spaceborne remote estimations of greenhouse gas atmospheric concentrations can offer the global coverage that is necessary to improve the constraint on their fluxes, thus enabling a better monitoring of anthropogenic emissions. In this work, we introduce the Adaptable 4A Inversion (5AI) inverse scheme that aims to retrieve geophysical parameters from any remote sensing observation. The algorithm is based on Bayesian optimal estimation relying on the Operational version of the Automatized Atmospheric Absorption Atlas (4A/OP) radiative transfer forward model along with the Gestion et Étude des Informations Spectroscopiques Atmosphériques: Management and Study of Atmospheric Spectroscopic Information (GEISA) spectroscopic database. Here, the 5AI scheme is applied to retrieve the column-averaged dry-air mole fraction of carbon dioxide ($X_{CO_2}$) from measurements performed by the Orbiting Carbon Observatory-2 (OCO-2) mission, and uses an empirically corrected absorption continuum in the $O_2$ A-band. For airmasses



below 3.0, $X_{CO_2}$ retrievals successfully capture the latitudinal variations of $CO_2$, as well as its seasonal cycle and long-term
increasing trend. Comparison with ground-based observations from the Total Carbon Column Observing Network (TCCON)
yields a difference of 1.33 ± 1.29 ppm, which is similar to the standard deviation of the Atmospheric $CO_2$ Observations from
Space (ACOS) official products. We show that the systematic differences between 5AI and ACOS results can be fully
removed by adding an average 'calculated – observed' spectral residual correction to OCO-2 measurements, thus underlying
the critical sensitivity of retrieval results to forward modelling. These comparisons show the reliability of 5AI as a Bayesian
optimal estimation implementation that is easily adaptable to any instrument designed to retrieve column-averaged dry-air
mole fractions of greenhouse gases.

## 1. Introduction

The atmospheric concentration of carbon dioxide ($CO_2$) has been rising for decades because of fossil fuel emissions as well
as land-use changes, and large uncertainties still remain in the global carbon budget (e.g. Le Quéré et al., 2009). In order to
address the global challenge of climate change, a better understanding of carbon sources and sinks is necessary and remote
spaceborne estimations of $CO_2$ columns can help constraining these carbon fluxes in atmospheric inversion studies, and thus
reducing the remaining uncertainties (e.g. Rayner and O'Brien, 2001; Chevallier et al., 2007; Basu et al., 2013, 2018).

The column-averaged dry-air mole fraction of $CO_2$ ($X_{CO_2}$) can be retrieved from thermal infrared (TIR) soundings, mostly
sensitive to the mid-troposphere (e.g. Chédin et al., 2003; Crevoisier et al., 2004, 2009a), as well as from near-infrared (NIR)
and shortwave infrared (SWIR) measurements, which are sensitive to the whole atmospheric column, and especially to levels
close to the surface, where carbon fluxes take place. The Scanning Imaging Absorption Spectrometer for Atmospheric
Chartography (SCIAMACHY) (Bovensmann et al., 1999) mission provided the first retrievals of $X_{CO_2}$ from NIR and SWIR
measurements with the Weighting Function Modified Differential Optical Absorption Spectroscopy (WFM-DOAS) least-
squares algorithm (Buchwitz et al., 2005). Bayesian Optimal Estimation, assuming a priori state and covariance for the
atmospheric parameters (Rodgers, 2000), was preferred for the more recent $X_{CO_2}$ retrieval algorithm Bremen Optimal
Estimation DOAS (BESD) dedicated to SCIAMACHY (Reuter et al., 2010, 2011). Current NIR and SWIR satellite missions
observing greenhouse gases include the Japanese Greenhouse gases Observing SATellites (GOSAT and GOSAT-2),
NASA's Orbiting Carbon Observatory-2 and 3 (OCO-2 and OCO-3), the Chinese mission TanSat and Sentinel 5-Precursor
from the European Space Agency (ESA). Over time, different algorithms based on various assumptions have been developed
to exploit their measurements and retrieve greenhouse gas concentrations. Those include the Japanese National Institute for
Environmental Studies (NIES) algorithm (Yokota et al., 2009; Yoshida et al., 2011, 2013), as well as the Atmospheric $CO_2$
Observations from Space (ACOS) algorithm (Bösch et al., 2006; Connor et al., 2008; O'Dell et al., 2012, 2018), UoL-FP
from the University of Leicester (Parker et al., 2011), RemoTeC from the Netherlands Institute for Space Research (SRON)



(Butz et al., 2011; Wu et al., 2018) and the Fast atmOspheric traCe gAs retrievaL (FOCAL) algorithm from the University of
Bremen (Reuter et al., 2017a, 2017b).

Besides implementing different inverse methods, these algorithms also rely on different forward radiative transfer models to
compute synthetic measurements and their partial derivatives. WFM-DOAS and BESD use SCIATRAN (Rozanov et al.,
2002, 2014) with the time-efficient correlated-k approximation (Buchwitz et al., 2000), and take into account multiple
scattering. The $X_{CO_2}$ retrievals performed by NIES (Yoshida et al., 2011) use a fast radiative transfer model that uses the k-
space to increase computational speed for multiple scattering (Duan et al., 2005). RemoTeC uses LINTRAN v2.0, which is a
linearized vector (handling the four components of the Stokes vector at the same time) radiative transfer forward model that
employs forward-adjoint theory to solve the radiative transfer equation (Hasekamp and Landgraf, 2002; Schepers et al.,
2014). The ACOS $X_{CO_2}$ retrieval algorithm and UoL-FP combine, in a piecemeal approach, the LIDORT model to perform a
scalar single-scattering radiative transfer computation with the discrete ordinate method (Spurr, 2002) and a second-order-of-
scattering polarization model named 2OS (Natraj et al., 2008). FOCAL uses a scalar radiative transfer model that
approximates multiple-scattering by assuming the presence of a unique optically thin isotropic scattering layer in the
atmosphere, thus enabling fast forward modelling (Reuter et al., 2017a).


These radiative transfer models also fundamentally depend on spectroscopic databases containing the parameters enabling to
compute the atmospheric gas absorption. The previously mentioned retrieval algorithms mainly rely on the HITRAN
spectroscopic database that evolved over the years: WFM-DOAS uses HITRAN 2008 (Rothman et al., 2009) as does the
UoL-FP (with some updated $CO_2$, $H_2O$ and $CH_4$ spectroscopic lines). RemoTeC and GOSAT $X_{CO_2}$ retrievals use HITRAN
2008 combined with an $O_2$ A-band line absorption spectroscopic model taking into account line-mixing and collision-
induced absorption (CIA) (Tran and Hartmann, 2008) as well as another line-mixing model for $CO_2$ lines (Lamouroux et al.,
2010). BESD relies on ABSCO v4.0 (computed by ACOS for OCO-2 processing), as does FOCAL for $H_2O$ (Thompson et
al., 2012, Reuter et al., 2017a, 2017b). Finally, the ACOS $X_{CO_2}$ retrieval algorithm producing the OCO-2 official product
uses ABSCO v5.0 (Drouin et al., 2017; O'Dell et al., 2018; Oyafuso et al., 2017), as does FOCAL, for $O_2$ and $CO_2$ (Reuter et
al., 2017a).

The design of an $X_{CO_2}$ retrieval algorithm, from the forward model and the spectroscopic parameters it uses to the choice of
the adjusted quantities in the state vector, has a critical influence on the overall performance of the observing system
(Rodgers, 2000). The systematic errors in retrieved $X_{CO_2}$ and their standard deviations (the latter being also called single
measurement precision) with regard to the true (but unknown) state of the atmosphere particularly impact the uncertainty
reduction and bias in atmospheric $CO_2$ flux inversion studies (e.g. Chevallier et al., 2007). However, direct in-situ
measurements of $CO_2$ atmospheric concentration profiles are logistically too difficult to scale up for systematic validation of





spaceborne measurements, and so retrieved $X_{CO_2}$ products are most often validated against columns with similar observation geometry, like the ground based solar absorption spectrometry. The Total Carbon Column Observing Network (TCCON) is a
network of ground stations that retrieve column-averaged dry-air mole fraction of $CO_2$ and other species from NIR and SWIR spectra measured with Fourier Transform Spectrometers (FTS) directly pointing at the sun (Wunch et al., 2011b). The network currently consists of 27 stations all around the world and its products constitute a "truth-proxy" reference for the validation of spaceborne retrievals of greenhouse gas atmospheric concentrations. For instance, TCCON datasets were used to validate SCIAMACHY (Reuter et al., 2011), GOSAT $X_{CO_2}$ retrieved by the ACOS (Wunch et al., 2011a) and NIES
algorithms (Inoue et al., 2016) and OCO-2 $X_{CO_2}$ produced by ACOS (O'Dell et al., 2018; Wunch et al., 2017), RemoTeC (Wu et al., 2018) and FOCAL (Reuter et al., 2017b). These three last algorithms exhibit different biases with regard to TCCON, depending on their respective forward modelling and bias correction strategies:  0.30 ± 1.04 ppm, 0.0 ± 1.36 ppm and 0.67 ± 1.34 ppm for OCO-2 nadir land soundings, respectively.

In this paper, we present the Adaptable 4A Inversion (5AI) that relies on the OPerational version of the Automatized Atmospheric Absorption Atlas (4A/OP) radiative transfer model (Scott and Chédin, 1981; Tournier, 1995; Cheruy et al., 1995) (https://4aop.aeris-data.fr) and the GEISA (Gestion et Étude des Informations Spectroscopiques Atmosphériques: Management and Study of Spectroscopic Information) spectroscopic database (Jacquinet-Husson et al., 2016) (http://cds-espri.ipsl.fr/etherTypo/?id=950). Here, version 2015 of GEISA is used. The 5AI scheme is applied to retrieve $X_{CO_2}$ from (1)
OCO-2 cloud-free target session soundings between 2014 and 2018 and (2) a sample of two years of OCO-2 nadir clear sky measurements with a global land coverage. We compare 5AI retrieval results to TCCON, and to ACOS and FOCAL v08 results over identical sets of soundings in order to assess the reliability of 5AI as a Bayesian optimal estimation implementation.

This paper is organized as follows: Sect. 2 describes the 5AI retrieval scheme and its current features, as well as the 4A/OP radiative transfer model, the GEISA spectroscopic database and the empirically corrected $O_2$ A-band absorption continuum on which it relies. Section 3 presents the OCO-2 and TCCON data selection. Section 4 presents the a posteriori filters used for this work and shows the 5AI $X_{CO_2}$ target and nadir retrieval results which are compared to TCCON, ACOS and available FOCAL v08 $X_{CO_2}$ products. Section 4 finally underlines the critical importance of forward modelling differences to explain
systematic differences between different $X_{CO_2}$ products through an average calculated – observed spectral residual correction. Section 5 highlights the conclusions of this work.



## 2. The 5AI retrieval scheme

As for any other retrieval scheme, 5AI aims at finding the estimate of atmospheric and surface parameters (for example trace gas concentration, temperature profile, surface albedo, or scattering particle optical depth) that best fits hyperspectral
measurements made from space. This inverse problem can be expressed with the following equation:

$$y = F(x) + \varepsilon \tag{1}$$

where $y$ is the measurement vector containing the radiances measured by the space instrument, $x$ is the state vector containing the geophysical parameters to be retrieved, $\varepsilon$ is the measurement noise and finally $F$ is the forward radiative transfer model that describes the physics linking the geophysical parameters to be retrieved to the measured infrared
radiances.

### 2.1 Forward modelling: 4A/OP and GEISA spectroscopic database

The 5AI retrieval scheme uses the OPerational version of the Automatized Atmospheric Absorption Atlas (4A/OP). 4A/OP is an accurate line-by-line radiative transfer model that enables a fast computation of atmospheric transmittances based on atlases containing pre-computed monochromatic optical thicknesses for reference atmospheres. Those are used to compute
atmospheric transmittances, for any input atmospheric profile and viewing configuration, that enable to solve the radiative transfer equation and yield radiances and their partial derivatives with regard to the input geophysical parameters at a pseudo-infinite spectral resolution (0.0005 cm$^{-1}$ best) or convolved with an instrument function. 4A/OP is the reference radiative transfer model for the Centre National d'Études Spatiales (CNES) / EUMETSAT IASI Level 1 Calibration/Validation and operational processing, and it is used for daily retrieval of mid-tropospheric columns of $CO_2$
(Crevoisier et al., 2009a) and $CH_4$ (Crevoisier et al., 2009b) from the Infrared Atmospheric Sounding Imager (IASI). Moreover, 4A/OP has also been chosen by CNES as the reference radiative transfer model for the development of the New Generation of the IASI instrument (IASI-NG) (Crevoisier et al., 2014).

Although originally developed for the thermal infrared spectral region, 4A/OP now also includes near and shortwave
infrared regions (NIR and SWIR). The extension to NIR and SWIR brought important new features to 4A/OP: (1) The computation of the atlases of optical thickness was extended to the 3,000 – 13,500 cm$^{-1}$ domain and takes into account line-mixing and CIA in the $O_2$ A-band (Tran and Hartmann, 2008) as well as line-mixing and $H_2O$-broadening of $CO_2$ lines (Lamouroux et al., 2010). The absorption lines of $CO_2$ we use in this work are thus identical to those included in HITRAN 2008; (2) Solar spectrum is a flexible input and the Doppler shift of its lines is computed; (3) The radiative transfer model is
now coupled with the LIDORT model (Spurr, 2002) for scalar multiple-scattering simulation performed with the discrete ordinates method, as well as with VLIDORT (Spurr, 2006) if polarization or Bidirectional Reflectance Distribution Functions (BRDF) need to be taken into account. These new features are critical for the preparation of the French NIR and



SWIR $CO_2$ remote sensing MicroCarb mission (Pascal et al., 2017) and the French-German MEthane Remote sensing LIdar Mission (MERLIN) (Ehret et al., 2017).


The 4A/OP radiative transfer model can be used with monochromatic optical thickness atlases computed from any spectroscopic database. For this present work, the atlases are computed using the GEISA 2015 (Gestion et Étude des Informations Spectroscopiques Atmosphériques: Management and Study of Spectroscopic Information) spectroscopic database. Being the base of many work since the beginning in the astronomical and astrophysical communities, GEISA has

been also used since the 2000's for the preparation of several current and future spatial missions, as to be chosen by CNES as the reference spectroscopic database for the definition of IASI-NG, MicroCarb and MERLIN. Due to imperfections in the Tran and Hartmann (2008) line mixing and CIA models, an empirical correction to the absorption continuum in the $O_2$ A-band, fitted from Park Falls TCCON spectra following the method described in Drouin et al. (2017), has been added. Finally, we use Toon (2015) as input solar spectra.

**2.2 Inverse modelling in the 5AI retrieval scheme**

**2.2.1 Bayesian optimal estimation applied for $X_{CO_2}$ retrieval**

The whole formalism of Bayesian optimal estimation that enables to find a satisfying solution to Eq. (1) may be found in Rodgers (2000). This subsection only outlines the key steps that are implemented in order to retrieve $X_{CO_2}$.

Equation (1) includes $\varepsilon$, the experimental noise of the measured radiances. Hence, it appears more appropriate to use a formalism that takes into account this measurement uncertainty and translates it into retrieval uncertainty. Considering the probability density function instead of vectors can bring such an insight. With Gaussian statistics, the inversion problem boils down to the minimization of the following $\chi^2$ cost function:

$$\chi^2 = \left(y - F(x)\right)^T S_e^{-1}\left(y - F(x)\right) + (x - x_a)^T S_a^{-1}(x - x_a) \tag{2}$$

where $x_a$ is the *a priori* state vector, which is also in most cases chosen as the first guess for iterative retrievals. Assuming again Gaussian statistics, $S_a$ is the *a priori* state covariance matrix that represents the variability around the *a priori* state vector, and similarly $S_e$ is the *a priori* measurement error covariance matrix that represents the noise model of the instrument. Moreover, as radiative transfer is a highly non-linear forward model, it is practical to use a local linear approximation, here expressed around the *a priori* state:

$$F(x) = F(x_a) + \frac{\partial F}{\partial x}(x_a)(x - x_a) . \tag{3}$$

The partial derivatives of the forward radiative transfer model $F$ (here 4A/OP) are expressed as a matrix, called the Jacobian matrix, and denoted $K$.





All these assumptions enable the maximum posterior probability state $\hat{x}$ that minimizes the cost function defined in Eq. (2)

to be found. It can be computed by iteration, using the general approach:

$$x_{i+1} = x_i + \left[(1 + \gamma)S_a^{-1} + K_i^T S_e^{-1} K_i\right]^{-1} \left(K_i^T S_e^{-1}(y - F(x_i)) - S_a^{-1}(x_i - x_a)\right) \tag{4}$$

where $\gamma$ is a scaling factor that can be set to 0 (Gauss-Newton method) or whose value can be adapted along iterations in order to prevent divergence (Levenberg-Marquardt method). $K_i$ denotes here the forward radiative transfer Jacobian matrix, whose values are evaluated for the state vector $x_i$. In this work we assume a slow variation of the Jacobian matrix along the

iterations and therefore choose not to update it in order to save computational time. Hence, the partial derivatives of the radiative transfer model are evaluated once and for all around the *a priori* state. We performed a sensitivity test and assessed that this approximation does not significantly change the retrieval results (not shown).

A successful retrieval reduces the *a priori* uncertainty of the state vector described in $S_a$. The *a posteriori* covariance matrix

of the retrieved state vector $\hat{S}$, whose diagonal elements give the posterior variance of the retrieved state vector elements, is expressed as

$$\hat{S} = [S_a^{-1} + K^T S_e^{-1} K]^{-1} \quad . \tag{5}$$

Finally, the sensitivity of the retrieval with regard to the true geophysical state $x_{true}$ is given by the averaging kernel matrix $A$ calculated according to

$$A = \frac{\partial \hat{x}}{\partial x_{true}}(x_a) = [S_a^{-1} + K^T S_e^{-1} K]^{-1} K^T S_e^{-1} K . \tag{6}$$

In most cases, the $CO_2$ concentration is included in the state vector as a level or layer profile from which $X_{CO_2}$, the retrieved column-averaged dry-air mole fraction of $CO_2$, is computed (e.g. O'Dell et al., 2012). If we note $\hat{x}_{CO2}$, the part of the retrieved state vector $\hat{x}$ containing the $CO_2$ profile, and $A_{CO2}$ and $\hat{S}_{CO2}$, the corresponding square parts of $A$ and $\hat{S}$, we have:

$$X_{CO_2} = h.\hat{x}_{CO_2} \tag{7}$$

$$\sigma_{X_{CO_2}} = \sqrt{h^T \hat{S}_{CO_2} h} \tag{8}$$

$$(a_{CO_2})_j = \frac{\partial X_{CO_2}}{\partial x_{true}} = \frac{\left(h^T A_{CO_2}\right)_j}{h_j} \tag{9}$$

where $h$ is the pressure weighting function. $\sigma_{X_{CO_2}}$ denotes the posterior uncertainty of the retrieved $X_{CO_2}$ and $a_{CO_2}$ is the $CO_2$ column averaging kernel. This profile vector describes the vertical sensitivity of the retrieved column with regard to the true

profile: it is essential to characterize retrieval results and to compare them to other products, as shown in Sect. 4.2.

### 2.2.2 5AI features and retrieval scheme setups for OCO-2

The 5AI retrieval scheme enables the retrieval of multiple geophysical variables from hyperspectral measurements. Those currently include trace gas concentration represented in the state vector as a concentration profile or a profile scaling-factor,




global temperature profile offset, surface temperature and pressure, band-wise albedo whose spectral dependence is

modelled as a polynomial, and finally scattering particle layer-wise optical depth.

For this work, the iterative scheme is set to the Levenberg-Marquardt method. The state vector includes the main geophysical parameters necessary to retrieve $X_{CO_2}$ and is described in Table 1. The a priori values and their covariance are identical to those used in the ACOS B8r version (O'Dell et al., 2018) in order to ease the retrieval result comparison, as we

aim to assess 5AI reliability. However, some elements of the ACOS state vector are not included in this work: scattering particles optical depth (AOD) as we only consider clear-sky soundings, Solar Induced Fluorescence which is not modelled in 4A/OP, surface wind speed (only land retrievals are considered) and Empirical Orthogonal Function (EOF) scaling factors.

**Table 1. 5AI state vector composition for OCO-2 retrievals**

| Variable name | Length | A priori value | A priori uncertainty (1$\sigma$) | Notes |
|---|---|---|---|---|
| $H_2O$ scaling factor | 1 | 1.0 | 0.5 (same as ACOS) | - |
| $CO_2$ layer concentration | 19 layers | ACOS a priori | ACOS prior covariance matrix | See prior covariance matrix in (O'Dell et al., 2012) |
| Surface Pressure | 1 | ACOS a priori | 4.0 hPa (same as ACOS) | - |
| Temperature profile offset | 1 | ACOS a priori | 5.0 K (same as ACOS) | - |
| Surface Albedo (order 0 of albedo model) | 3 bands | ACOS a priori | 1.0 (same as ACOS) | Evaluated at 0.77, 1.615 and 2.06 µm for $O_2$, $CO_2$ weak and strong bands, respectively |
| Surface Albedo Slope (order 1 of albedo model) | 3 bands | 0.0 | 1.0 /cm$^{-1}$ | (O'Dell et al., 2018) explains that this uncertainty is 0.0005 /cm$^{-1}$ but B8r data release uses 1.0 /cm$^{-1}$ in the 'apriori_covariance_matrix', in 'RetrievalResults', in Diagnostics files. |


4A/OP is used with VLIDORT for $O_2$ A-band polarized forward computations, and the ACOS Stokes coefficients are applied to yield the final scalar radiances. For $CO_2$ weak and strong bands, scattering and polarization can be neglected in clear sky conditions, and only the Stokes coefficient 0.5 for the *I* component of the electric field is applied to yield the final scalar radiances.



**3. Data**

**3.1 Data description**

The OCO-2 spectrometer measures Earth-reflected near and shortwave infrared (NIR and SWIR) sunlight in three distinct bands: the $O_2$ A-band (0.7 µm), the weak $CO_2$ band (1.6 µm) and the strong $CO_2$ band (2.0 µm). The satellite has three distinct observation modes. The nadir and glint modes are the nominal science observation modes; they constitute the vast

majority of OCO-2 measurements. In addition, the target mode of the OCO-2 mission provides data for the validation of the retrievals. During a target session, the satellite tilts and aims at a validation target (most of them are TCCON stations) and scans its whereabouts several times during the overpass. These sessions thus provide with OCO-2 data points closely collocated with validation targets (over areas that can be as small as 0.2° longitude × 0.2° latitude) and registered over a few minutes (Wunch et al., 2017).


OCO-2 high-resolution spectra are analysed by the ACOS team in order to retrieve $X_{CO_2}$ and other geophysical parameters from them. Two different $X_{CO_2}$ values are provided by the ACOS team: raw and posterior bias-corrected $X_{CO_2}$. Raw $X_{CO_2}$ is the direct output of the ACOS algorithm following the full physics retrieval: its most recent version is distributed within the B8 retrospective (B8r) ACOS data release (O'Dell et al., 2018). Posterior bias-corrected $X_{CO_2}$ is an empirically corrected

$X_{CO_2}$ that has reduced averaged bias with regard to different "truth-proxies" (O'Dell et al., 2018). The last available version of this product is distributed within the B9 retrospective (B9r) ACOS data release. It corrects the impacts of footprint geolocation errors and erroneous prior surface pressure temporal sampling directly in the bias correction procedure applied to the B8r raw $X_{CO_2}$ product, without a complete full-physics reprocessing of all OCO-2 data (Kiel et al., 2019). In this work, 5AI results are compared with B8r raw $X_{CO_2}$, and B9r posterior bias-corrected $X_{CO_2}$ are also shown.


In addition to ACOS products, we also compare our results with OCO-2 FOCAL v08 data produced at the University of Bremen with the FOCAL algorithm (Reuter et al., 2017a) that includes an empirical posterior bias correction directly on the top of the full-physics retrieval (Reuter et al., 2017b). Only the posterior bias-corrected $X_{CO_2}$ is included in FOCAL v08 data.

In this work, we compare $X_{CO_2}$ retrieved from OCO-2 spectra to TCCON data. The TCCON network uses ground-based high resolution Fourier Transform Spectrometers to measure NIR and SWIR spectra that enable the retrieval of the column-averaged dry-air mole fractions of greenhouse gases. These retrievals are performed by GGG2014 (Wunch et al., 2015) and their results are available on the TCCON Data Archive (https://tccondata.org/).



### 3.2 Data selection

We intend to compare 5AI results with regard to TCCON against ACOS and FOCAL results for corresponding sets of soundings. First, we select all the OCO-2 target soundings between 2015 and 2018 with low ACOS retrieved total AOD (<0.5) and ACOS cloud, sounding quality and outcome flags at their best possible value. As FOCAL v08 uses prior and posterior filtering techniques that are different from ACOS, only a fraction of this first selection intersects with available FOCAL data. In order to increase this fraction, we add all OCO-2 points with the best ACOS cloud and sounding quality

flags intersecting the available FOCAL v08 data points, whatever ACOS outcome flag and retrieved AOD. This composite sample set includes 48,885 OCO-2 target soundings and the fraction of available intersecting FOCAL data is shown in Fig. 1.

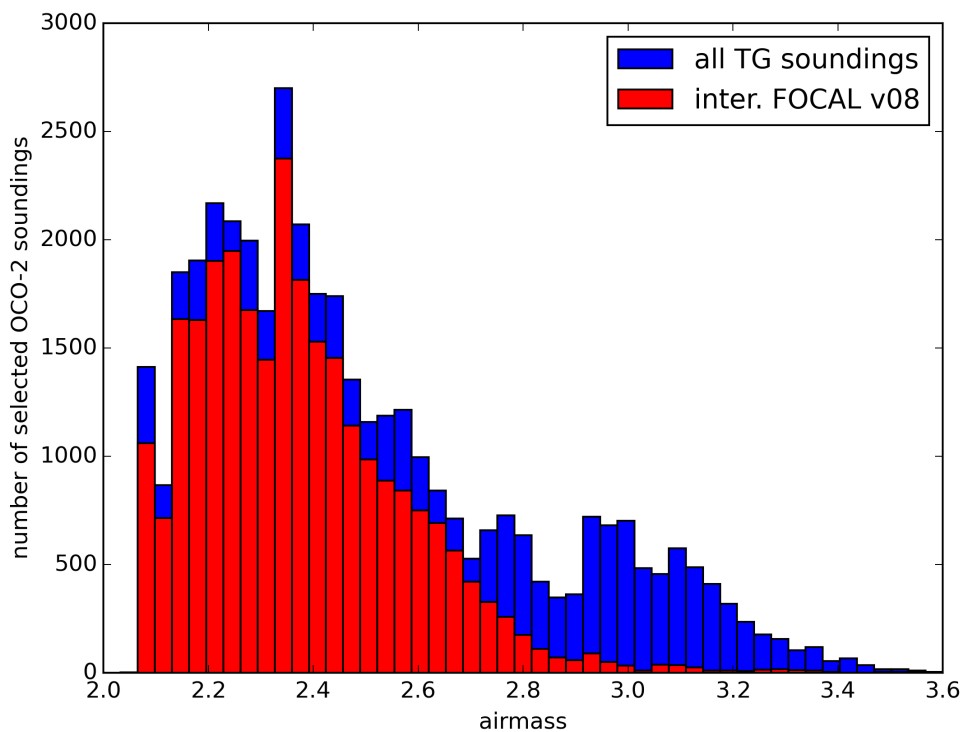

**Figure 1.** Airmass distributions of all the selected OCO-2 target soundings (blue) and of those intersecting available FOCAL v08
data (red).





For this study, we select the TCCON official products measured ± 2 hours with regard to OCO-2 overpass time and only keep the target sessions where at least five OCO-2 measurements passing 5AI posterior filters and five TCCON data points are available. This set includes 11,102 TCCON individual retrieval results from 20 TCCON stations listed in Table 2.


**Table 2. TCCON data used in this work**

| TCCON station | Coordinates (latitude, longitude, altitude) | Number of target sessions | Date range (first and last sessions) | Reference |
|---|---|---|---|---|
| Ascension Island | 7.92S, 14.33W, 0.01 km | 4 | 2015-01-16 – 2018-01-15 | (Feist et al., 2014) |
| Bialystok (Poland) | 53.23N, 23.03E, 0.18 km | 1 | 2015-03-18 | (Deutscher et al., 2019) |
| Bremen (Germany) | 53.10N, 8.85E, 0.027 km | 1 | 2016-03-17 | (Notholt et al., 2014) |
| Burgos (Philippines) | 18.53N, 120.65E, 0.035 km | 2 | 2017-04-21 – 2018-03-07 | (Morino et al., 2018a) |
| Caltech (USA) | 34.14N, 118.13W, 0.230 km | 21 | 2014-09-12 – 2018-09-16 | (Wennberg et al., 2015) |
| Darwin (Australia) | 12.424S, 130.89E, 0.03 km | 8 | 2015-05-15 – 2017-07-28 | (Griffith et al., 2014a) |
| Edwards (USA) | 34.96N, 117.88W, 0.700 km | 3 | 2015-07-04 – 2018-08-22 | (Iraci et al., 2016) |
| Eureka (Canada) | 80.05N, 86.42W, 0.61 km | 2 | 2015-06-16 – 2015-06-28 | (Strong et al., 2019) |
| Izana (Tenerife) | 28.31N, 16.50W, 2.37 km | 2 | 2018-01-05 – 2018-03-24 | (Blumenstock et al., 2017) |
| Karlsruhe (Germany) | 49.10N, 8.44E, 0.116 km | 3 | 2016-05-07 – 2017-07-06 | (Hase et al., 2015) |
| Lamont (USA) | 36.60N, 97.49W, 0.32 km | 12 | 2015-02-10 – 2016-11-11 | (Wennberg et al., 2016) |
| Lauder (New Zealand) | 45.04S, 169.68E, 0.37 km | 4 | 2015-02-17 – 2017-01-30 | (Sherlock et al., 2014) |
| Orléans (France) | 47.97N, 2.11E, 0.13 km | 2 | 2015-04-08 – | (Warneke et al., 2019) |





| | | | | | |
|---|---|---|---|---|---|
| | | | | 2018-06-26 | |
| Paris (France) | 48.85N, 2.36E, 0.06 km | 1 | | 2016-08-25 | (Té et al., 2014) |
| Park Falls (USA) | 45.95N, 90.27W, 0.44 km | 7 | | 2014-10-11 – 2017-04-21 | (Wennberg et al., 2017) |
| Réunion Island | 20.90S, 55.49E, 0.087 km | 4 | | 2015-03-24 – 2015-08-01 | (De Mazière et al., 2017) |
| Saga (Japan) | 33.24N, 130.29E, 0.007 km | 6 | | 2015-07-31 – 2018-03-10 | (Kawakami et al., 2014) |
| Sodankylä (Finland) | 67.37N, 26.63E, 0.188 km | 4 | | 2015-08-20 – 2018-07-17 | (Kivi et al., 2014; Kivi and Heikkinen, 2016) |
| Tsukuba (Japan) | 36.05N, 140.12E, 0.03 km | 6 | | 2014-11-14 – 2017-06-17 | (Morino et al., 2018b) |
| Wollongong (Australia) | 34.40S, 150.88E, 0.03 km | 13 | | 2014-09-23 – 2018-05-06 | (Griffith et al., 2014b) |

Besides target sessions, we also select a sample of clear sky OCO-2 nadir land soundings with a coverage as global as possible over the years 2016-2017 (all ACOS flags at their best value possible). For every month and 5° longitude × 5°

latitude bins we select 25 (10 for North-America, South-Africa and Australia) soundings with low ACOS retrieved total AOD. For 2016 and 2017, this selection is done for a maximum ACOS retrieved total AOD of 0.035 and 0.045, respectively, yielding 17,069 soundings for 2016 and 11,002 for 2017. Figure 2 shows the spatial and temporal distribution of these OCO-2 points.





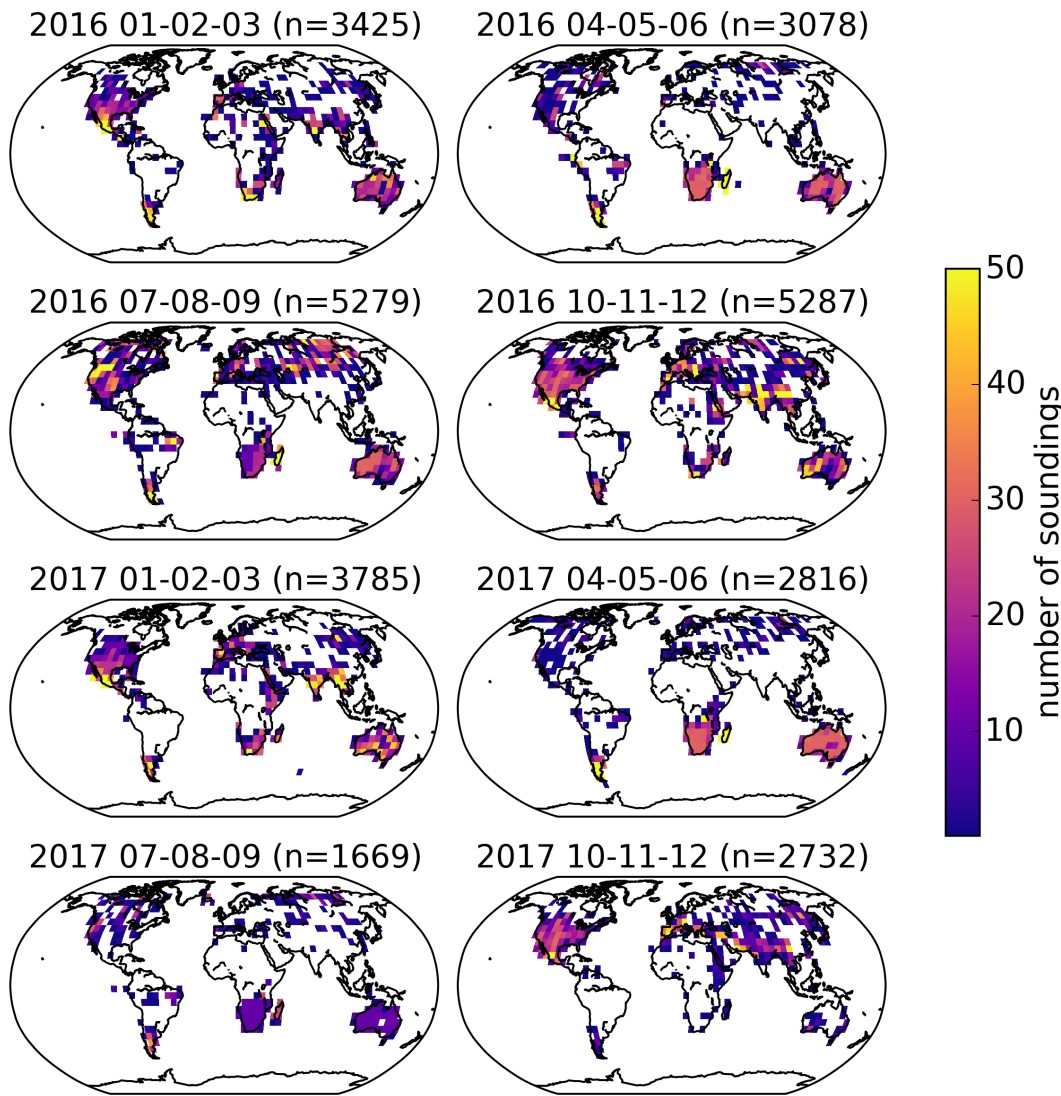

Figure 2. Spatial and temporal repartition of the sample of nadir OCO-2 soundings selected for 5AI retrievals, in seasonal and 5°
× 5° square bins. The titles include the number of soundings n for the corresponding panel: the low number of selected soundings
in July-August-September 2017 is due to an identified OCO-2 data gap.





## 4. Results and discussion

### 4.1 Post-filtering of retrieval results

We apply the a posteriori filters described in Table 3 to ensure retrieval results' quality. The surface pressure filter removes soundings for which it proved difficult to successfully model the optical path, suggesting scattering related errors leading to a large difference between the retrieved and prior surface pressure. The reduced $\chi^2$ filter removes the worst spectral fits. In the end, 88% of our selected soundings pass these first two filters. In addition, the blended albedo filter removes the fraction

of target data (29%) representative of challenging snow or ice-covered surfaces (Wunch et al., 2011a). With the current retrieval setup, the difference between the 5AI retrieved surface pressure and its prior exhibit an airmass dependence as shown in Fig. 3. For this present work, we filter out all sounding with airmasses above 3.0. Future studies will refine the 5AI forward and inverse setup in order to process hyperspectral infrared soundings with larger airmasses. Results detailed in the following subsections are based on the 24,449 target and 21,254 nadir OCO-2 soundings that passed all these filters.


Table 3. Filters applied on 5AI retrieval results for this work.

| Variable name | Minimum value | Maximum value | Definition and reference | OCO-2 mode |
|---|---|---|---|---|
| Retrieved surface pressure | $P_{nlev-1}$ | - | The atmosphere is discretized in 20 levels bounding 19 layers. We do not allow the surface pressure, $P_{nlev}$, to be lower than its preceding pressure level. | Nadir, Target |
| Reduced $\chi^2$ | - | 7.0 | Overall goodness of the spectral fit (e.g. Wu et al., 2018) | Nadir, Target |
| Blended albedo | - | 0.8 | 2.4 x $O_2$ A-band albedo + 1.13 x $CO_2$ strong band albedo (Wunch et al., 2011a, 2017) | Target |
| Airmass | - | 3.0 | $\frac{1}{\cos(SZA)} + \frac{1}{\cos(VZA)}$, with SZA, the solar zenith angle, and VZA, the viewing zenith angle (Wunch et al., 2011a) | Nadir, Target |



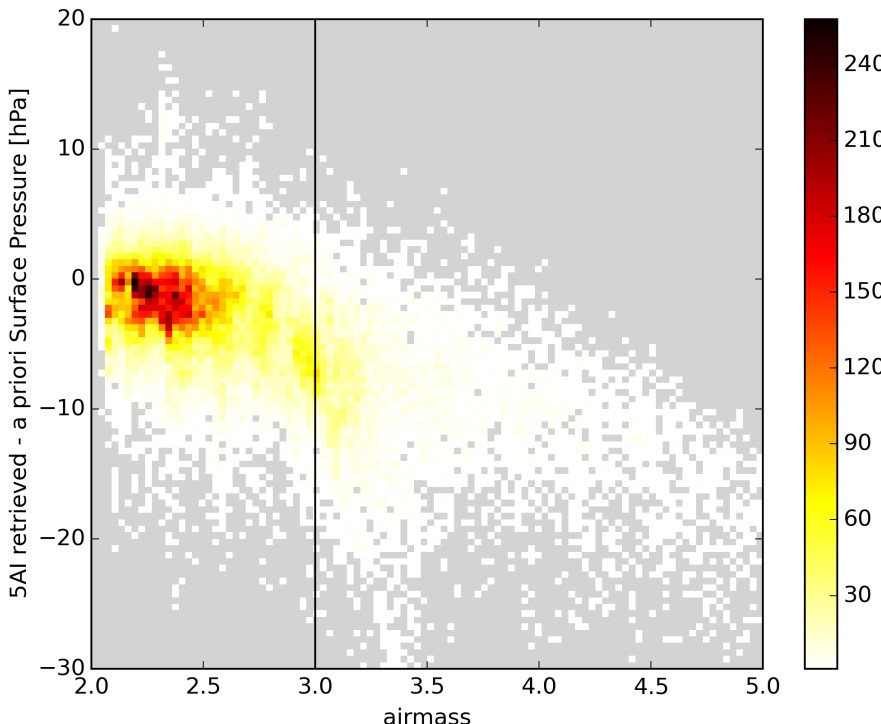

**Figure 3.** Distribution of target and nadir 5AI retrievals passing surface pressure, blended albedo and reduced $\chi_r^2$ filters
according to airmass and difference between retrieved and prior surface pressures. Grey areas denote bins for which no 5AI
retrieval is available.

### 4.2 OCO-2 target retrieval results

For every target session, we consider a unique average of the available retrieval results from OCO-2 measurements and a

unique average of the corresponding TCCON official products as performed in e.g. O'Dell et al. (2018) and Wu et al.

(2018). As OCO-2 and TCCON $X_{CO_2}$ vertical sensitivities described by their averaging kernels are not exactly identical, we

take into account the averaging kernel correction of TCCON data as performed by the ACOS team (O'Dell et al., 2018) and

described by Eq. (10) (Nguyen et al., 2014):

$$X_{OCO-2,TCCON} = X_{a\,priori} + (\frac{\hat{X}_{TCCON}}{X_{a\,priori}} - 1) \sum_j h_j (a_{CO_2})_j x_{a\,priori,j} .$$
(10)



$X_{OCO-2,TCCON}$ is the column-averaged dry-air mole fraction of $CO_2$ that would have been retrieved from the OCO-2
measurement if the collocated TCCON retrieval was the true state of the atmosphere, $X_{a\ priori}$, the a priori column-averaged
dry-air mole fraction of $CO_2$, considered to be very similar between 5AI (or ACOS) and GGG2014, $\hat{X}_{TCCON}$, the TCCON
retrieved column-averaged dry-air mole fraction of $CO_2$, $\boldsymbol{h}$, the pressure weighting function vector defined previously,
$(\boldsymbol{a_{CO_2}})$, the $CO_2$ column averaging kernel vector defined in Eq. (9) and $\boldsymbol{x_{a\ priori}}$, the a priori $CO_2$ concentration profile
vector. The effect of this correction yields a positive shift of the bias with regard to TCCON of about 0.2 ppm for the set of
target sessions considered in this work.

Following post-filtering, Fig. 4 shows 5AI raw results compared to the TCCON official product over 106 target sessions.
The mean systematic $X_{CO_2}$ bias (5AI $-$ TCCON) is 1.33 ppm and its standard deviation is 1.29 ppm. The ACOS raw $X_{CO_2}$
and TCCON $X_{CO_2}$ comparison for the corresponding set of OCO-2 soundings is also presented in Fig. 4: the bias with regard
to TCCON is -2.08 ppm and its standard deviation is 1.27 ppm. This difference in bias compared to TCCON may be greatly
influenced by forward modelling differences between 5AI and ACOS, as detailed later in this work. Bias-corrected
RemoTeC $X_{CO_2}$ retrieval results compared to the ACOS official product exhibit similar differences in bias standard
deviations (Wu et al., 2018).

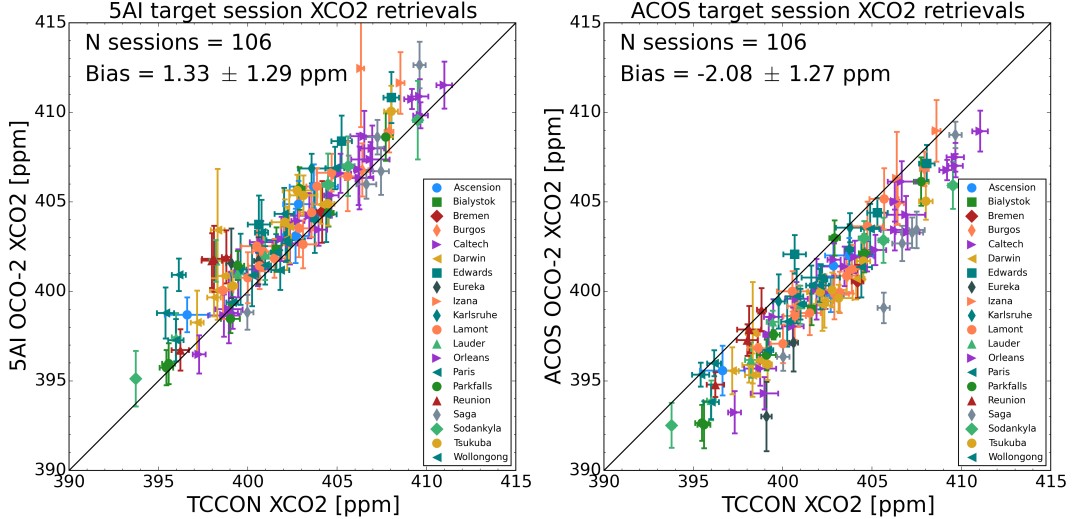

**Figure 4. 5AI (left) and raw ACOS B8r (right) OCO-2 target $X_{CO_2}$ retrieval results compared to TCCON official $X_{CO_2}$ product.**
**Individual sounding results are averaged for every target session: markers show session average for OCO-2 and TCCON $X_{CO_2}$,**
**and error bars show standard deviations.**





Temporal and latitudinal fits of 5AI and ACOS $X_{CO_2}$ biases compared to TCCON are displayed in Fig. 5. Temporal biases
are fitted with a 1st order polynomial added to a cosine and exhibit quasi-null slope with a ~0.4 ppm amplitude of yearly
oscillation in both 5AI and ACOS cases. Latitudinal bias fits performed with all the available target sessions except those
from Eureka (full lines) show that 5AI bias compared to TCCON appears to be larger in the Southern hemisphere than in the
Northern hemisphere, but its behaviour is quite parallel to ACOS except at higher latitudes where 5AI and ACOS get closer.

The Eureka station (latitude 80°N) has been removed from those fits as satellite retrievals and validation are known to be
challenging at these latitudes (O'Dell et al., 2018). The same latitudinal bias fits performed on the dataset intersecting
available FOCAL v08 data (dashed lines) show improved 5AI bias compared to TCCON. This is mainly due to the airmass
distribution difference between the two sets displayed in Fig. 1.

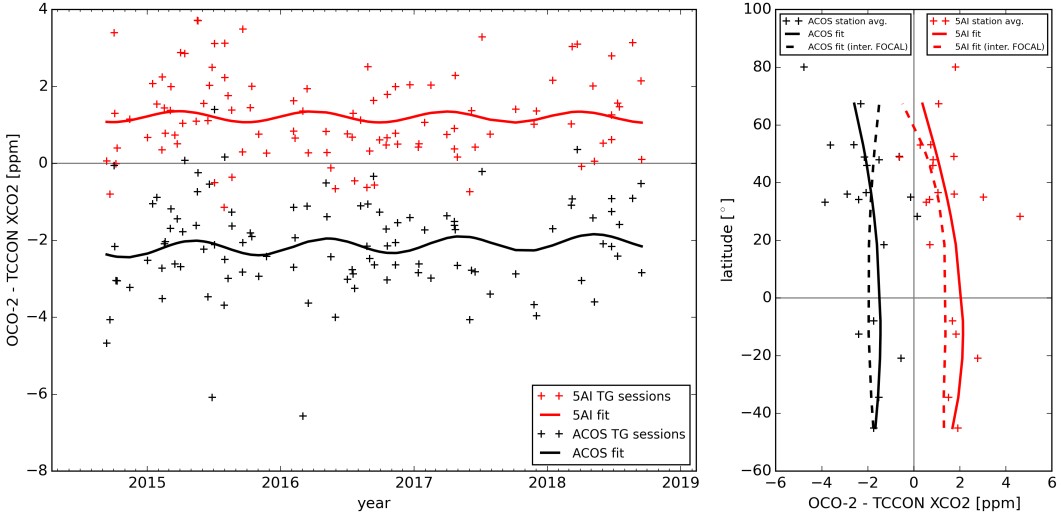

**Figure 5. 5AI and raw ACOS B8r OCO-2 target $X_{CO_2}$ bias with regard to TCCON as a function of time (left panel) and latitude
(right panel). Crosses show individual session averages in the left panel and individual station averages in the right panel, full lines
show polynomial fits of this bias for all target sessions, and dashed lines represent the polynomial fits of this bias for the target
sessions intersecting FOCAL v08 available soundings, used for the simplistic empirical bias correction applied in Fig. 6.**

Finally, a consistent comparison of 5AI, ACOS and FOCAL v08 on this intersecting set of available soundings is performed
in Fig. 6. Its first column shows 5AI and ACOS raw $X_{CO_2}$ results. As previously mentioned, FOCAL v08 only distributes a
posterior bias-corrected $X_{CO_2}$ product. Thus, in order to provide a more consistent comparison of the three retrieval schemes,
in the second column of Fig. 6, we apply a simplistic empirical correction on 5AI and ACOS results that removes the fitted
latitudinal bias with regard to TCCON, presented in dashed lines in the right panel of Fig. 5. Finally, the last column of Fig.





6 shows official posterior bias-corrected ACOS B9r and FOCAL v08 products. The standard deviations of these biases are
quite similar between the three retrieval schemes (0.05 ppm difference between 5AI and ACOS, 0.01 ppm difference
between 5AI and FOCAL v08). The slight improvement of 5AI bias compared to TCCON between Fig. 4 and Fig. 6 is due
to the differences in airmass distribution between the two sounding sets.

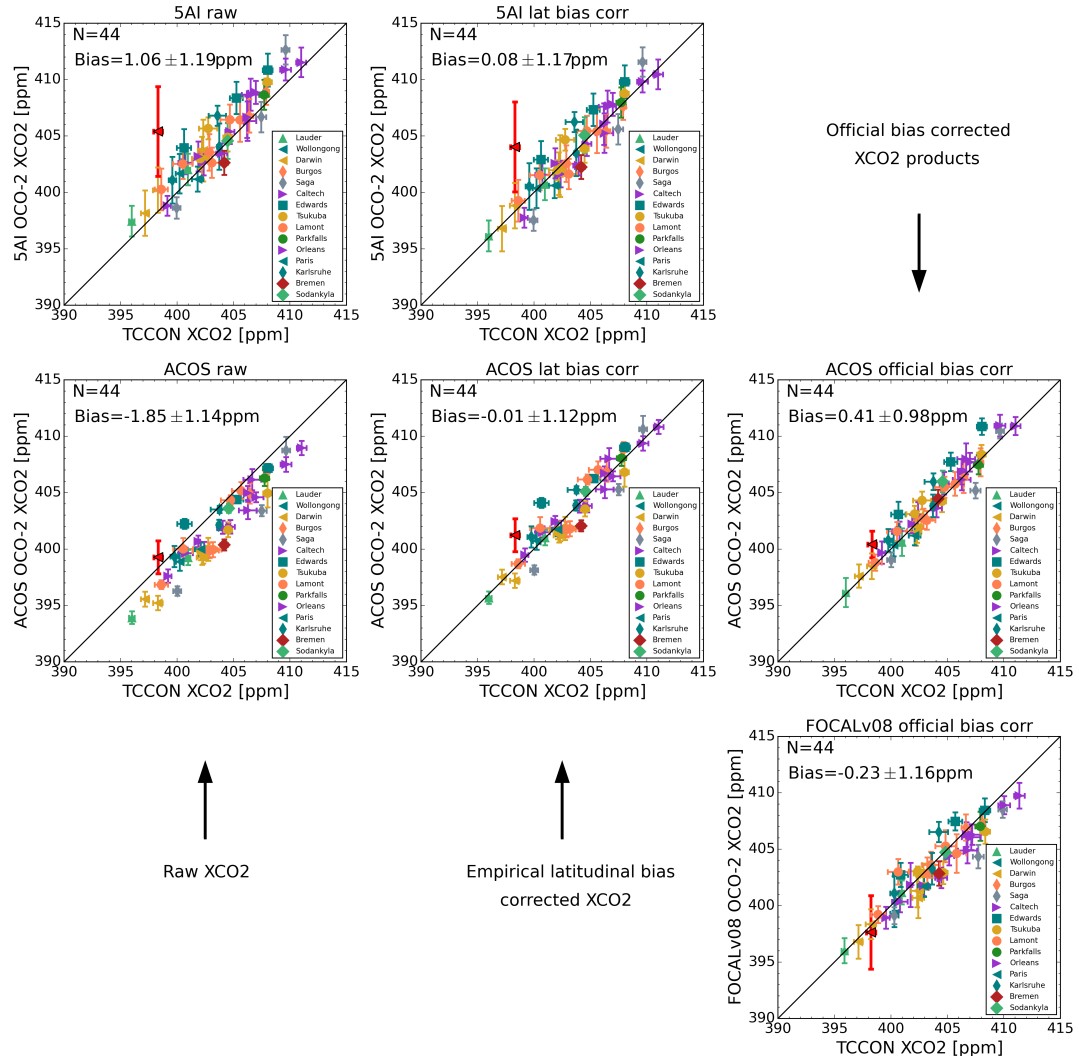

**Figure 6. 5AI (top row panels), ACOS B8r for raw products and B9r for the official bias-corrected one (center row panels) and**





FOCAL v08 (bottom row panel) OCO-2 target $X_{CO_2}$ retrieval results compared to TCCON official $X_{CO_2}$ product. Depending on data availability, we show raw $X_{CO_2}$ results (left column panels), simplistically corrected $X_{CO_2}$ results based on a latitudinal bias fit (central column panels) and official bias-corrected $X_{CO_2}$ products (right column panels). Individual sounding results are averaged for every target session: markers show session average for OCO-2 and TCCON $X_{CO_2}$, and error bars show standard deviations. One target session in Darwin on the 11[th] of September 2015 distinguishes itself from other sessions with either increased bias compared to TCCON or OCO-2 session-wise standard deviation for the three algorithms. It has been manually removed from the statistics but still appears in red with black lining in the figure.

**4.3 OCO-2 nadir retrieval results**

In this subsection, raw 5AI retrieved $X_{CO_2}$ is compared to the ACOS raw product on a sample of OCO-2 nadir clear sky soundings as described in Sect. 3 and displayed in Fig. 2. The nadir viewing configuration is the nominal science mode of the OCO-2 mission and allows comparisons at a larger spatial scale than the one offered by the target mode dedicated to validation.

Figure 7 shows the average and associated standard deviation of the difference between 5AI and ACOS retrieved raw $X_{CO_2}$. The overall 5AI-ACOS difference is about 3 ppm, with a latitudinal dependency: it is lower above mid-latitudes in the Northern hemisphere. The standard deviation is mainly correlated with topography: it is higher in the vicinity of mountain chains and lower on flatter areas. As we do not take into account topography in the sampling strategy of the processed OCO-2 nadir soundings, its greater variability in mountainous areas can result in a greater variability of the retrieved surface pressure which is strongly correlated with retrieved $X_{CO_2}$. As for the highest standard deviations in South America, they may be caused by the South Atlantic Anomaly to which they are close (Crisp et al., 2017).

**Figure 7.** Spatial repartition of 5AI – raw ACOS B8r average difference and its standard deviation on 5° × 5° square bins for the nadir data selection.






As seen in Fig. 8, latitudinal variations of raw 5AI retrieved $X_{CO_2}$ are consistent with those of ACOS, with a difference between the two products almost constant except above mid-latitudes in the Northern hemisphere where the differences are smaller. In addition, the comparison between 5AI and ACOS in nadir mode is consistent with the results obtained for target sessions. Indeed, the raw 5AI – ACOS target difference lies within $\pm\ 1\ \sigma$ of nadir results, with $\sigma$ the standard deviation of

the 5AI – ACOS difference. Figure 9 details the temporal variations of the retrieved $X_{CO_2}$. The global long-term increase of the atmospheric concentration of $CO_2$ can be observed in both hemispheres as well as the seasonal cycle, stronger in the Northern hemisphere where most of the vegetation respiration and photosynthesis happen. The temporal variations of the 5AI – ACOS $X_{CO_2}$ retrieval differences in nadir mode are also consistent with those presented in target mode.



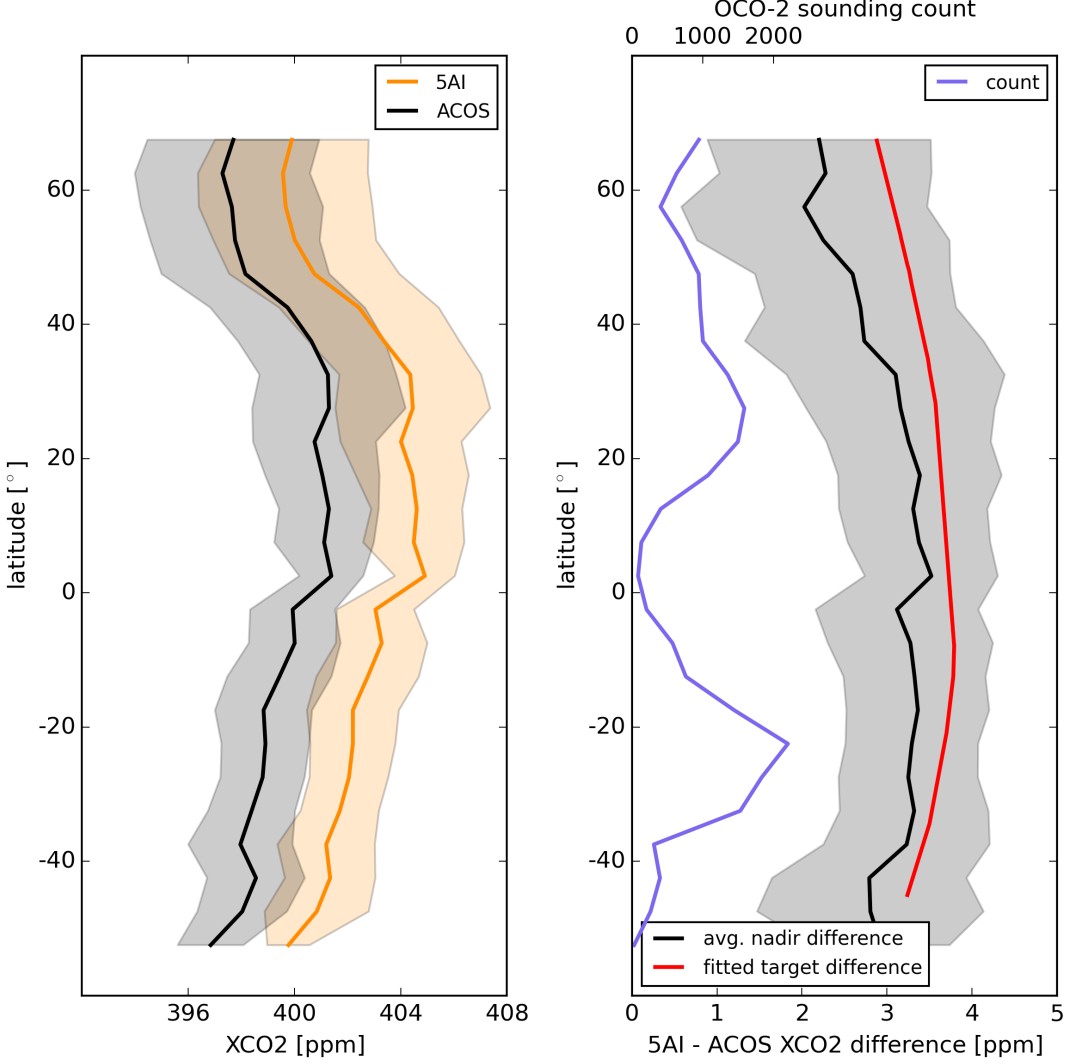

**Figure 8. Latitudinal variation of 5AI and raw ACOS B8r retrieved $X_{CO_2}$ (left) and their difference (right). The right panel compares 5AI-ACOS average difference for nadir soundings and 5AI-ACOS difference fitted on target sessions (bottom axis). The number of available nadir soundings is also shown in the right panel (top axis).**

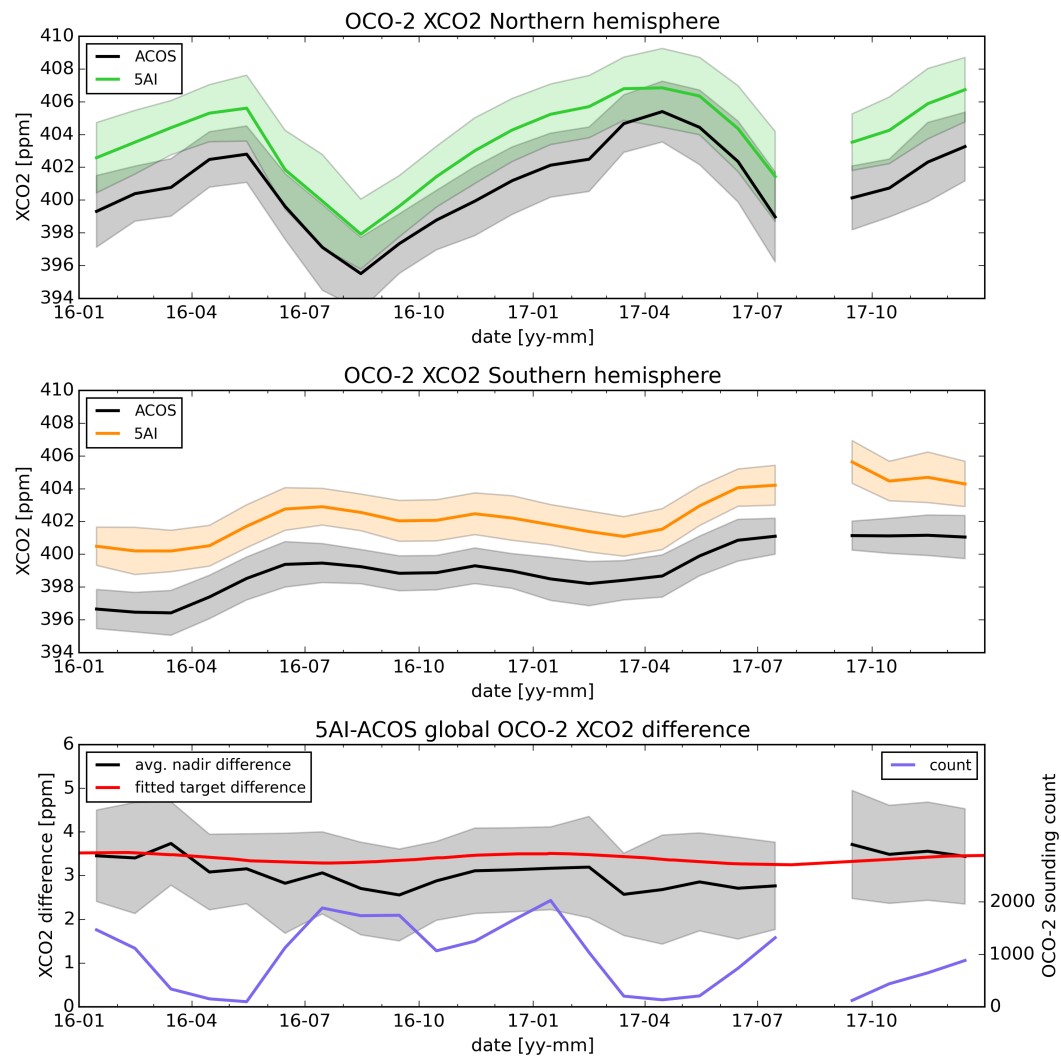

**Figure 9. Temporal variation of 5AI and raw ACOS B8r retrieved $X_{CO_2}$ in the Northern hemisphere (top), Southern hemisphere (center) and the global difference (bottom). The bottom panel compares 5AI-ACOS difference for nadir and target OCO-2.**






**4.4 Sensitivity of raw retrieval results to forward modelling**

A difference of about 3 ppm is found between 5AI and ACOS raw $X_{CO_2}$ retrieved from OCO-2 for both nadir and target observations. As mentioned in Sect. 1 and 2, 5AI and ACOS retrieval schemes rely on different radiative transfer models and

spectroscopic inputs, and their respective retrieval setups are also quite different. In order to quantify the impact of these differences, we perform an average 'calculated – observed' spectral residual analysis (hereafter 'calc – obs'), where the calculated spectrum (convolved to OCO-2 Instrument Line Shape) is generated by the forward model 4A/OP using GEISA spectroscopic database and the ACOS retrieval results (posterior pressure grid, temperature, $H_2O$ and $CO_2$ profiles as well as albedo and albedo slope), and is compared to the corresponding OCO-2 observation. In addition, possible background

differences are compensated by scaling the OCO-2 spectrum so that its transparent spectral windows fit those of the 4A/OP calculated spectrum. This comparison is performed for a randomly chosen half of the nadir OCO-2 points with an airmass below 3.0 selected in 2016 (6,790 in total). Figure 10 shows the resulting averaged calculated – observed spectral residuals as well as the typical transmission of the OCO-2 measurements. Differences are principally located in the 0.7 µm $O_2$ absorption band, but also in the 1.6 and 2.0 µm $CO_2$ absorption bands. They are due to the radiative transfer models'

differences between ACOS and 5AI (parametrization of continua, spectroscopy, etc).



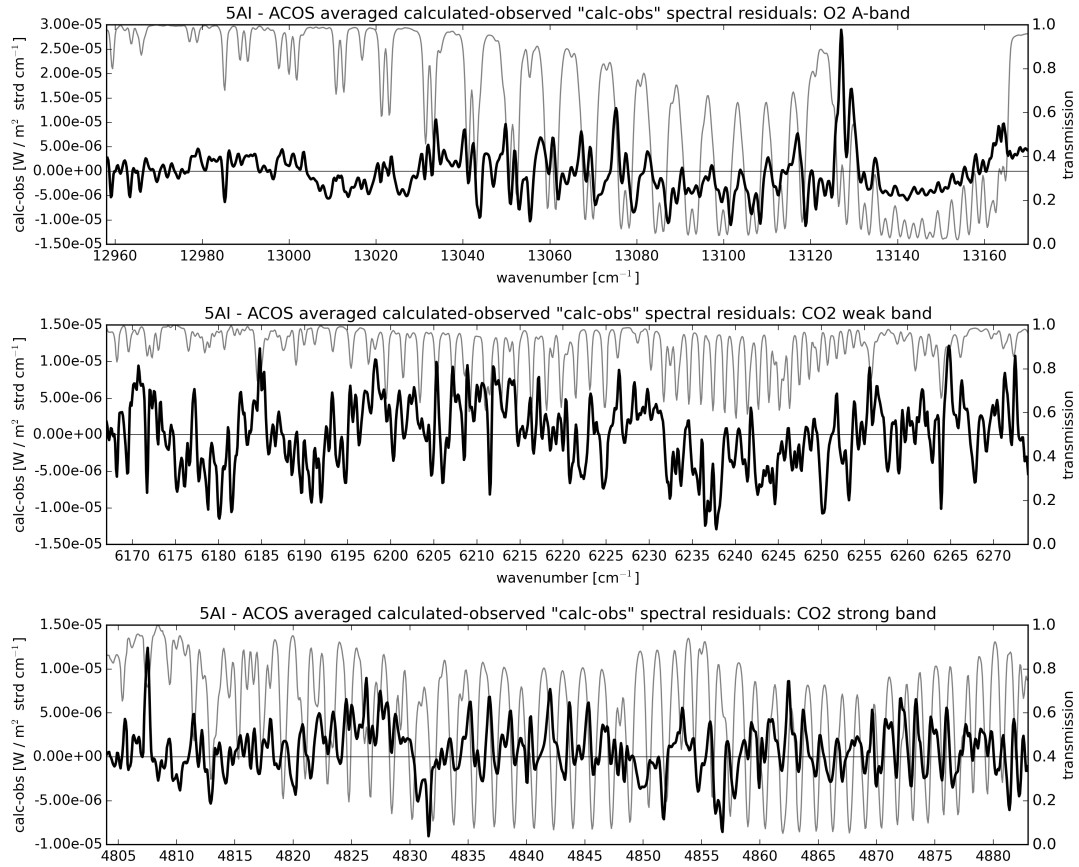

**Figure 10. 5AI – ACOS average calculated – observed 'calc-obs' spectral residuals in the O$_2$ A-band (top panel), CO$_2$ weak band (middle panel) and CO$_2$ strong band (bottom panel) appear in thick black lines (left axis). Typical transmissions for the three bands are shown in thin grey lines (right axis).**


In order to compare 5AI retrievals with ACOS products while attenuating the impact of the forward modelling differences, the obtained averaged calc − obs residual is added to every OCO-2 measurements within the complementary half of 2016 selected nadir soundings (6,799 in total) to compensate for the systematic radiative model differences between 4A/OP and

ACOS. We then apply the 5AI inverse scheme on this new dataset. Figure 11 compares the distributions of 5AI − ACOS retrieval results obtained with and without the calc − obs adjustment. The systematic differences between 5AI and ACOS





results for $X_{H_2O}$, $X_{CO_2}$, surface pressure and global temperature profile shift are fully removed when adding the spectral residual adjustment to OCO-2 measurements. This allows a first quantification of how spectroscopic and radiative transfer differences can impact $X_{CO_2}$ retrievals. This calc – obs adjustment impacts the standard deviations of 5AI – ACOS

differences. Indeed, several retrieval setup and forward modelling differences such as scattering particle parameters remain unaccounted for in this analysis. Their impact may be attenuated by the background difference correction, which, if disabled, leads to a similar standard deviation of 5AI – ACOS differences in both with and without calc – obs cases. However, without the background compensation, the average difference between 5AI – ACOS is only reduced to 1.9 ppm for $X_{CO_2}$ (not shown). This exemplifies how highly challenging the sounding-to-sounding inter-comparison of retrieval results remains,

and highlights how forward modelling and retrieval setup design impact $X_{CO_2}$ retrieval results.





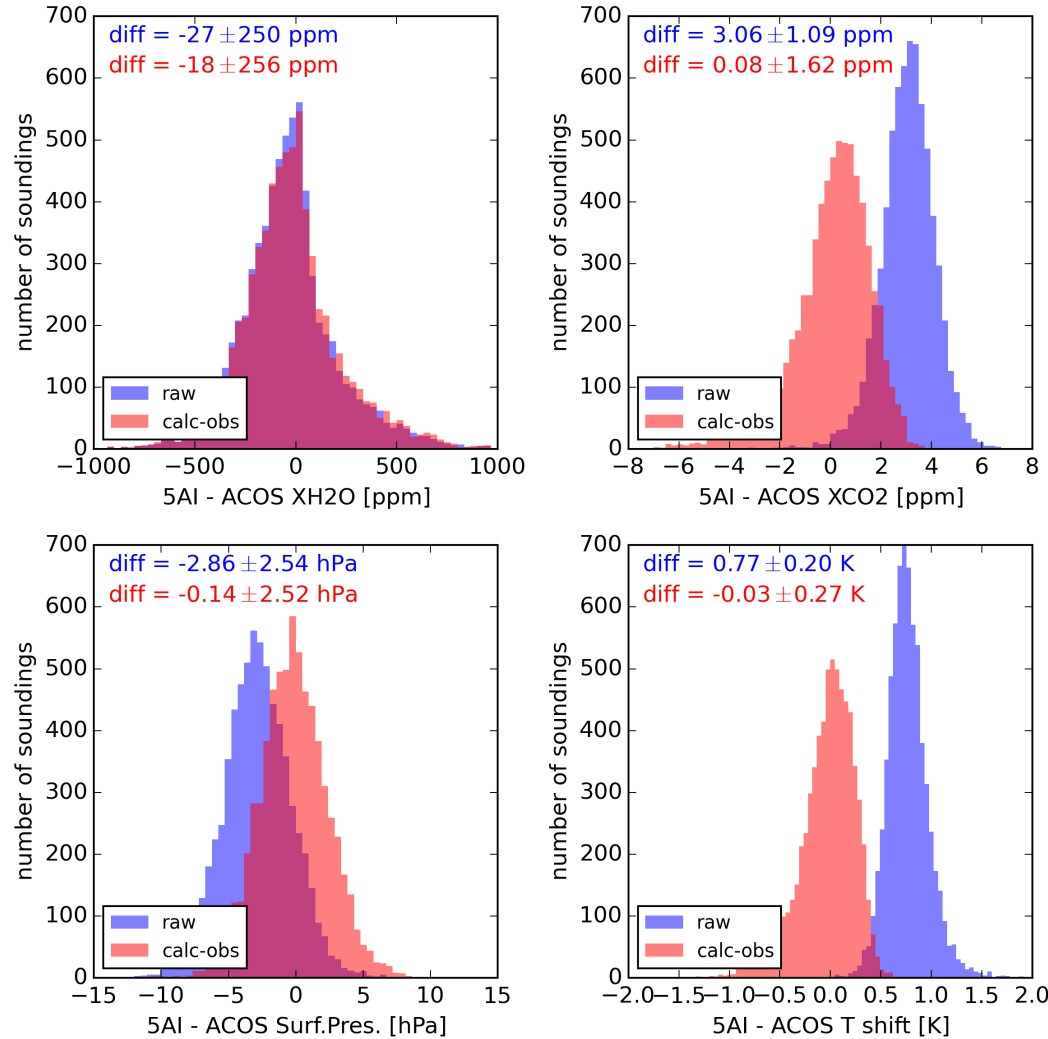

**Figure 11.** 5AI - raw ACOS B8r difference distributions for $X_{H_2O}$ (top left), $X_{CO_2}$ (top right), surface pressure (bottom left) and temperature profile global shift (bottom right) showed without applying the average calc-obs spectral residual correction (in blue) and with the correction (in red).




## 5. Conclusions

In this work, we have introduced the 5AI inverse scheme: it implements Bayesian optimal estimation and uses the 4A/OP radiative transfer model with the GEISA spectroscopic database and an empirically corrected absorption continuum in the $O_2$ A-band. We have applied the 5AI inverse scheme to retrieve $X_{CO_2}$ from a sample of ~77k OCO-2 clear-sky soundings with

low ACOS retrieved total AOD in target and nadir mode. Its global averaged uncorrected bias with regard to TCCON is 1.33 ppm with a standard deviation of 1.29 ppm for airmasses below 3.0. These results are comparable in standard deviation with those obtained by ACOS and FOCAL v08 for corresponding sets of OCO-2 soundings. Moreover, we showed that, similarly to ACOS, 5AI $X_{CO_2}$ retrievals satisfactorily capture the global increasing trend of atmospheric $CO_2$, its seasonal cycle as well as its latitudinal variations, and that 5AI results are consistent between OCO-2 nadir and target modes. Although 5AI

exhibits a difference of ~3 ppm with regard to ACOS, we showed that forward modelling differences between 5AI and ACOS can be removed with an average 'calculated – observed' spectral residual correction added to OCO-2 measurements, thus underlying the critical sensitivity of retrieval results to forward modelling.

For favourable conditions (clear sky, low ACOS total AOD), we showed that 5AI is a reliable implementation of the optimal

estimation algorithm whose results can be compared to other available products. Efforts are underway in order to optimize and increase the speed of 4A/OP coupling with LIDORT and VLIDORT, and hence to process more soundings and account for cirrus clouds or aerosols in the retrievals. Additionally, 5AI retrieval setup will be refined to process soundings with airmasses larger than 3.0 in future works. Finally, the implementation of the 5AI retrieval scheme is intended to be compatible with 4A/OP structure, so that the code can be easily adapted to any current or future greenhouse gas monitoring

instrument, from TCCON or EM27/SUN (e.g. Gisi et al., 2012; Hase et al., 2016) to OCO-2, MicroCarb (Pascal et al., 2017) or $CO_2$ Monitoring (Meijer and Team, 2019), and even applied to research concepts such as the one proposed in the European Commission H2020 SCARBO project (Brooker, 2018).

### Data availability

For this work we use the B8r and B9r releases of OCO-2 data that were produced by the OCO-2 project at the Jet Propulsion

Laboratory, California Institute of Technology, and obtained from the OCO-2 data archive maintained at the NASA Goddard Earth Science Data and Information Services Center (NASA GES-DISC). TCCON data are available on the TCCON Data Archive (https://tccondata.org/) and FOCAL v08 data can be downloaded on the FOCAL-OCO2 website hosted by the University of Bremen (http://www.iup.uni-bremen.de/~mreuter/focal.php). 5AI retrieval results presented in this work are available upon request from Matthieu Dogniaux by email (matthieu.dogniaux@lmd.ipsl.fr).





**Competing interests**

The authors declare that they have no conflict of interest.

**Acknowledgements**

This work has received funding from CNES and CNRS. M. Dogniaux is funded by Airbus Defence and Space in the
framework of a scientific collaboration with École polytechnique. The authors would like to thank the ACOS team and
NASA for OCO-2 data availability as well as Maximilian Reuter and the University of Bremen for FOCAL v08 data
availability.

The TCCON site at Réunion Island is operated by the Royal Belgian Institute for Space Aeronomy with financial support
since 2014 by the EU project ICOS-Inwire and the ministerial decree for ICOS (FR/35/IC1 to FR/35/C5) and local activities
supported by LACy/UMR8105 – Université de La Réunion. The TCCON stations at Tsukuba and Burgos are supported in
part by the GOSAT series project. Local support for Burgos is provided by the Energy Development Corporation (EDC,
Philippines). The Paris TCCON site has received funding from Sorbonne Université, the French research center CNRS, the
French space agency CNES, and Région Île-de-France. The Ascension Island TCCON station has been supported by the
European Space Agency (ESA) under grant 4000120088/17/I-EF and by the German Bundesministerium für Wirtschaft und
Energie (BMWi) under grants 50EE1711C and 50EE1711E. We thank the ESA Ariane Tracking Station at North East Bay,
Ascension Island, for hosting and local support. N. Deutscher is funded by ARC Future Fellowship FT180100327. Darwin
and Wollongong TCCON stations are supported by ARC grants DP160100598, LE0668470, DP140101552, DP110103118
and DP0879468 and NASA grants NAG5-12247 and NNG05-GD07G.

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
