# Peer review of "The Adaptable 4A Inversion (5AI): Description and first XCO2 retrievals from OCO-2 observations"

_Atmospheric Measurement Techniques, 2020_

## Referee Comment (RC1) · Anonymous Referee #1 · 10 Dec 2020

**General comments:**

The manuscript entitled, "*The Adaptable 4A Inversion (5AI): Description and first XCO2 retrievals from OCO-2 observations*" presents a description of the 5AI retrieval, designed for use with OCO-2 but adaptable to other current and future GHG satellites. They show that, although there is a small offset of a few ppm, 5AI agrees in many ways with the B8/B9 ACOS XCO2 retrieval. The manuscript is very well-written and I recommend publication in AMT after the authors address comments below.

The primary weakness I see is that this is a non-scattering retrieval, which isn't mentioned until page 8 of the manuscript. This is important to discuss and likely contributes to the especially large differences seen between 5AI and ACOS in Africa, South America, India, etc. (Fig. 7) and the general lack of data in typically aerosol- or cloud-laden areas (Fig. 2). All other major near-infrared XCO2 retrieval algorithms include a scattering component because no scene is truly "clear sky" and you'll end up with unacceptably high biases unless you heavily filter the data. Annoyingly, the places we care about most regarding the carbon cycle are also cloudy and full of aerosols, so a retrieval needs to be able to at least get quality XCO2 for slightly contaminated scenes.

**Specific comments:**

- Maybe too many details in the introduction. E.g. listing all the HITRAN/ABSCO versions. Suggest moving elsewhere.

- P2 L64: S5P doesn't measure XCO2, so maybe not relevant here.

- P4 L121: which version of ACOS? B10 is the current version.

- "*In this work we assume a slow variation of the Jacobian matrix along the 200 iterations and therefore choose not to update it in order to save computational time… We performed a sensitivity test and assessed that this approximation does not significantly change the retrieval results (not shown).*"

Is this because not solving for a scattering component makes the retrieval much more linear?

- "*(O'Dell et al., 2018) explains that this uncertainty is 0.0005 /cm-1 but B8r data release uses 1.0 /cm-1 in the 'apriori_covariance_matrix', in 'RetrievalResults', in Diagnostics files.*"

Appears to be a typo in O'Dell 2018. 1.0 is correct for B8r.

- "*its most recent version is distributed within the B8 retrospective (B8r) ACOS data release*"

B10 is the latest version, as of a few months ago. But B8/B9 is fine for an analysis like this.

- *"we apply a simplistic empirical correction on 5AI"*

Have you thought about what you'll do for a more complex bias correction in the future?

- *"0.05 ppm difference between 5AI and ACOS"*

Are you comparing 5AI lat bias corr to ACOS lat bias corr? Don't you want to compare 5AI lat bias corr to ACOS official bias corr (so, 1.17 – 0.98, not 1.17 – 1.12)?

- *"and account for cirrus clouds or aerosols in the retrievals."*

This is critical. Figure 7 clearly shows the disadvantages of a non-scattering retrieval. Your algorithm differs substantially from ACOS where there are high levels of dust (e.g. Sahara), pollution (e.g. India), etc. And probably is suffering from an inability to do anything about unscreened thin clouds in general.

**Technical comments:**

P4 L120: "target mode" instead of "target session"
P6 L169: "many projects" instead of "many work". The sentence is a bit clunky.
P6 L188: Would recommend something like: "Moreover, as the forward model for this retrieval is highly non-linear…"

---

## Referee Comment (RC2) · Anonymous Referee #2 · 29 Dec 2020

The paper of Dogniaux et al. reports on the development and testing of a carbon dioxide retrieval algorithm (5AI) for spectroscopic solar backscatter measurements such as those conducted by the OCO-2 satellite. The algorithm is deployed and evaluated for an evaluation data set from the OCO-2 mission. The paper is well written and the analyses are sound. But the serious drawback of the study is that the proposed algorithm does not account for particle scattering i.e. the algorithm is not (yet) what is typically called a full-physics algorithm. And, the algorithm is also not built as a computationally efficient approximation, for which lower accuracy would be acceptable.

The neglect of particle scattering essentially reduces the retrieval problem to a trans-

mittance calculation which will induce substantial errors even if the particle load in the atmosphere is low (e.g. AOD <0.5, termed "clear-sky" by the study). I actually wonder why a sophisticated radiative transfer model such as (V)LIDORT is required at all to perform such calculations. Isn't it just Beer-Lambert's law? Maybe, molecular Rayleigh scattering is included?

For example, the differences in spectral residuals between 5AI and ACOS (illustrated in Fig. 10) most likely stem to a large part from the differences in particle treatment, which is corroborated by the finding that 5AI and ACOS retrievals become more similar if the spectral differences are synthetically added to the 5AI processing (section 4.4). Essentially, the ACOS particle treatment is "added" to the 5AI calculations.

Given that the remote sensing community has been working on the simultaneous retrievals of greenhouse gases and particle properties for many years, the study lags behind current developments. On the other hand, the paper appears to be one of the first presenting a new algorithm to be applied to the problem and, it is always scientifically interesting and important to compare to new approaches.

If to be published, the authors should clearly discuss the drawback of neglecting particle scattering and they should underline that this assumption is most likely the leading error term. In particular, they should mention it in prominent places such as abstract and conclusions and they should add a section where they describe the (lack of) scattering treatment. Currently, the manuscript reads like (V)LIDORT (with standard scattering treatment) is used and only quite late it becomes clear that particle scattering is neglected.

I understand that full-physics retrievals are tough to get to work, but the authors might want to consider to support their evaluation by some refined assessments wrt. particle scattering:

- For the analysis in section 4, it might be insightful to examine correlation plots of the 5AI-ACOS (and/or 5AI-TCCON) differences with scattering and radiative transfer

parameters (e.g. aerosol optical thickness retrieved by ACOS, surface albedo).

- The authors could add a forward modelling sensitivity study where they assume a particle scattering scenario (e.g. the one suggested by the ACOS retrievals) for which they actually perform full-physics forward calculations (without the derivatives) and then compare the spectral residuals to clear-sky calculations. Such an assessment could serve as a means to single out the effect of neglecting particle scattering in a clean way.

- ACOS (and other algorithms) previously reported on systematic challenges. For example, there was a bias in earlier ACOS retrievals towards the Southern higher-latitudes which was attributed to stratospheric aerosols (O'Dell et al., 2018). There was a land-ocean bias in early RemoTeC/GOSAT retrievals (Basu et al., 2013). For a future refined assessment, it would be interesting to evaluate 5AI in terms of these specific findings. [O'Dell, C. W, https://doi.org/10.5194/amt-11-6539-2018, 2018, Basu et al., https://doi.org/10.5194/acp-13-8695-2013, 2013.]

Other comments

General: The manuscript repeatedly emphasizes that the 5AI retrieval is a "Bayesian" concept. Is there more to it than just using the Bayesian/optimal estimation formalism (as most regularization concepts do)? A truly Bayesian retrieval would require a careful setup of the prior covariance (to represent the true covariance). Table 1 suggests that the prior covariance is rather a reasonable ad-hoc choice than the true atmospheric covariance. For example, 4 hPa surface pressure uncertainty is certainly a tight constraint compared to the actual atmospheric variability. Choosing prior covariances ad-hoc is common practice and thus, I do not criticize the procedure per se, but I do recommend being humble when highlighting the Bayesian and "optimal" nature of retrievals under such assumptions.

Introduction: The introductory discussion of other algorithms includes too much of detail. Many of the details (e.g. spectroscopic data, covariance setups, radiative transfer

speed-ups) evolve over time without being published. Some key conceptual aspects are missing e.g. the fact that RemoTeC has chosen to not retrieve surface pressure because the problem becomes too ill-posed with respect to the microphysical particle parameters. I would recommend summarizing some of the algorithm features to avoid the paper being already outdated when it gets published. Some important early conceptual work by Oshchepkov et al., 2008, could be added. [Oshchepkov et al., https://doi.org/10.1029/2008JD010061, 2008]

Equation (1): "F" should be bold face, since it is a vector. The comment applies to many occurrences of "F" that follow.

L180: "Epsilon" should be bold face, since it is a vector.

L181: "Considering the probability density function instead of vectors" What does the statement mean? Wouldn't it be more appropriate to highlight the key idea of Bayesian statistics? For example: "Assuming that the true state is a particular draw from a Gaussian statistics and that it can be found through a trade-off to a measurement constraint weighted by the respective uncertainties ..." (or better wording).

Equation (3): Notation "$(x\_a)$" is a bit misleading. It refers to the fact that the derivatives are evaluated at $x\_a$ (for the first iteration, if $x\_a$ is also the initial guess) while the following parentheses "$(x-x\_a)$" refers to a multiplication. I would recommend using a vertical line with subscript $x\_a$ to indicate the linearization point of the derivatives.

L198: The Levenberg-Marquardt parameter "gamma" is required to vanish at the final iteration since otherwise it will induce a (probably undesired) regularization of the problem. In my experience, not all readers are aware of this effect and thus, I would recommend mentioning it clearly.

L199: It would be interesting to quantify the advantage in computational cost when not updating the derivatives. Probably, not updating the derivatives means that convergence is somewhat slower and thus, more iterations are required. So, there is a

trade-off between number of iterations and cost per iteration. Could you give rough numbers?

Equation (6): A downside of not updating the Jacobians is that the averaging kernels are calculated with respect to the prior state (not the iterated retrieved state). The prior state is typically "further away" from the true state than the retrieval and thus, it is in the non-linear regime. I think this is a minor issue in general, but it might cause confusion when, for example, evaluating the effects of non-linearity on retrievals. One might consider calculating the Jacobians for the first and the last iteration to get rid of this effect.

L263: For a retrieval method paper, comparisons to raw retrievals of other algorithms are the essential tool. Comparisons to bias-corrected retrievals will not inform on methodological issues, but, to a large extent, such comparisons will mirror the effects of the bias corrections, in particular since some algorithms require substantial bias corrections. So, the (anyway sparse) comparisons to FOCAL could be removed from the manuscript.

Fig. 10. I recommend plotting the residuals and transmittances in the same units (either absolute radiance or relative transmittance) to allow for comparison of the amplitude of the residuals wrt. the spectra.

---

## Author Comment (AC1) · 22 Mar 2021

**We would like to thank the referee for their feedback and relevant comments. We will address every point in blue, between the referee's comments.**

**General comments:**

The manuscript entitled, "*The Adaptable 4A Inversion (5AI): Description and first XCO2 retrievals from OCO-2 observations*" presents a description of the 5AI retrieval, designed for use with OCO-2 but adaptable to other current and future GHG satellites. They show that, although there is a small offset of a few ppm, 5AI agrees in many ways with the B8/B9 ACOS XCO2 retrieval. The manuscript is very well-written and I recommend publication in AMT after the authors address comments below.

The primary weakness I see is that this is a non-scattering retrieval, which isn't mentioned until page 8 of the manuscript. This is important to discuss and likely contributes to the especially large differences seen between 5AI and ACOS in Africa, South America, India, etc. (Fig. 7) and the general lack of data in typically aerosol- or cloud-laden areas (Fig. 2). All other major near- infrared XCO2 retrieval algorithms include a scattering component because no scene is truly "clear sky" and you'll end up with unacceptably high biases unless you heavily filter the data. Annoyingly, the places we care about most regarding the carbon cycle are also cloudy and full of aerosols, so a retrieval needs to be able to at least get quality XCO2 for slightly contaminated scenes.

We thank both reviewers for stressing the importance of this discussion that was missing in the first submitted manuscript and gave a false impression regarding the capability of 5AI to include scattering parameters in its state vector. In the revised version, we have included 5AI retrievals that take into account scattering particle parameters in the state vector for a sub-sample of the selected OCO-2 target data, and we now discuss the impact on 5AI results and how they compare to ACOS in a dedicated subsection (Sect 5.1).

As 5AI does enable to take into account the impact of scattering particles in XCO2 retrievals, we took the opportunity of this necessary discussion to perform XCO2 retrievals while taking into account scattering particles, and thus try to assess the forward and inverse consequences of our initial hypothesis. As XCO2 retrievals take longer when considering the impact of scattering particles, especially for OCO-2 that requires using the coupling with VLIDORT, we only processed a few hundreds OCO-2 measurements of our target sounding selection. Because we are interested here in the impact of scattering particles, we focused on 15 target sessions that have collocated TCCON, OCO-2 and also available AERONET data. The independent AERONET information regarding scattering particle optical depth can thus help to discuss the retrieved total aerosol optical depth.

For these retrievals, we took into account two layers of aerosols: a coarse mode layer and a fine mode layer for which we added their respective optical depths in the 5AI state vector. We compared these new 5AI retrieval results to those obtained without considering scattering particles and identified several impacts:

1- reduction or even removal of the 5AI surface pressure airmass dependence, that can be explained by forward and inverse modeling arguments (see Fig. 2, Fig. 8)
2- shift in the averaged 5AI retrieved surface pressure, compared to the prior (see Fig. 8). This partly translates into an averaged XCO2 difference to ACOS that is reduced when taking into account scattering particles (see Fig. 9).

Both 5AI and ACOS retrieved optical depths show a large scatters compared to AERONET data (0.07 and 0.05, respectively).

Regarding the revised manuscript, we separated the Results (Sect. 4) and Discussion (Sect. 5) sections. Subsection 5.1 gives all the details regarding the discussion of the impacts of neglecting scattering particles in 5AI XCO2 retrievals. Of course, the inverse setup choices made for the sub-sample of 5AI retrievals that consider the impact of scattering particles are not exactly identical to ACOS. Differences remain and result in remaining systematic average differences between 5AI and AOCS. We thus kept the averaged – calculated spectral residual discussion, detailed in subsection 5.2, that enables to show that those systematic differences can be compensated.

**Specific comments:**

- Maybe too many details in the introduction. E.g. listing all the HITRAN/ABSCO versions. Suggest moving elsewhere.

Our intention was to underline the multiplicity of approaches that could be designed to retrieve XCO2 from infrared spectra, from the choice of inverse method, state vector setups, forward model choices and speed-ups and spectroscopic database.

We have adapted the introduction to reflect this great diversity of methods, design and spectroscopy choices without enumerating all of them.

- P2 L64: S5P doesn't measure XCO2, so maybe not relevant here.

We restricted to carbon dioxide observing instruments in the introduction (lines 61-63)

- P4 L121: which version of ACOS? B10 is the current version.

We use version 8 of the Full-Physics ACOS results. We added the version number here (lines 115-116), and repeated it in the Data section (lines 244-245)

- *"In this work we assume a slow variation of the Jacobian matrix along the iterations and therefore choose not to update it in order to save computational time... We performed a sensitivity test and assessed that this approximation does not significantly change the retrieval results (not shown)."* Is this because not solving for a scattering component makes the retrieval much more linear?

That is right. Trying to estimate XCO2 while taking into account scattering particles makes the retrieval way less linear and keeping the 1st Jacobian matrix in this case leads to unrealistic results. However, when we neglect the impact of scattering particles, as it is the case here (but for Sect. 5.1), the retrieval problem is more linear, making it possible to only work with the Jacobian matrix computed for the a priori state.

- *"(O'Dell et al., 2018) explains that this uncertainty is 0.0005 /cm-1 but B8r data release uses*

*1.0 /cm-1 in the 'apriori_covariance_matrix', in 'RetrievalResults', in Diagnostics files."*

Appears to be a typo in O'Dell 2018. 1.0 is correct for B8r.

Thank you for confirming this uncertainty value, we removed this comment from the revised manuscript.

- *"its most recent version is distributed within the B8 retrospective (B8r) ACOS data release"* B10 is the

latest version, as of a few months ago. But B8/B9 is fine for an analysis like this.

We updated the text that was written just before summer 2020 (lines 244-245).

- "*we apply a simplistic empirical correction on 5AI*"

Have you thought about what you'll do for a more complex bias correction in the future?

An operational large scale processing of OCO-2 data is out of the scope of this paper. To reach this goal, the question of empirical bias correction would be seriously considered, and different approaches would need to be investigated. The sole purpose of this simplistic empirical bias correction was to try to be more consistent when comparing to FOCAL. Following the advice of referee #2, we have removed this sparse and less consistent comparison from the revised manuscript.

- "*0.05 ppm difference between 5AI and ACOS*"

Are you comparing 5AI lat bias corr to ACOS lat bias corr? Don't you want to compare 5AI lat bias corr to ACOS official bias corr (so, 1.17 – 0.98, not 1.17 – 1.12)?

For this case, intersecting with available FOCAL data, we indeed compared lat bias corr results, because it appeared to be the less inconsistent comparison, with a standard deviation difference also similar to 5AI and ACOS raw. Comparing 5AI lat bias with the official ACOS bias corr appears less consistent as the simplistic lat bias correction did not have the ambition of an operational one such as the official ACOS bias correction. Following referee #2 advice we have removed this discussion.

- "*and account for cirrus clouds or aerosols in the retrievals.*"

This is critical. Figure 7 clearly shows the disadvantages of a non-scattering retrieval. Your algorithm differs substantially from ACOS where there are high levels of dust (e.g. Sahara), pollution (e.g. India), etc. And probably is suffering from an inability to do anything about unscreened thin clouds in general

 We reflected this comment when describing results of Fig. 7 (now Fig. 5 in the revised manuscript).

**Technical comments:**

P4 L120: "target mode" instead of "target session" P6 L169: "many projects" instead of "many work". The sentence is a bit clunky. P6 L188: Would recommend something like: "Moreover, as the forward model for this retrieval is highly non-linear..."

We have corrected these points in the revised manuscript as per your suggestions:

- "OCO-2 best flag target mode soundings between 2014 and 2018" (line 106)
- "Being the base of many projects since the beginning in the astronomical" (line 155)
- "Moreover, as the forward model for this retrieval is highly non-linear, it is practical to use a local linear, approximation, here expressed around the a priori state" (line 176-177)

---

## Author Comment (AC2) · 22 Mar 2021

The paper of Dogniaux et al. reports on the development and testing of a carbon dioxide retrieval algorithm (5AI) for spectroscopic solar backscatter measurements such as those conducted by the OCO-2 satellite. The algorithm is deployed and evaluated for an evaluation data set from the OCO-2 mission. The paper is well written and the analyses are sound. But the serious drawback of the study is that the proposed algorithm does not account for particle scattering i.e. the algorithm is not (yet) what is typically called a full-physics algorithm. And, the algorithm is also not built as a computationally efficient approximation, for which lower accuracy would be acceptable.

The neglect of particle scattering essentially reduces the retrieval problem to a transmittance calculation which will induce substantial errors even if the particle load in the atmosphere is low (e.g. AOD <0.5, termed "clear-sky" by the study).

All selected nadir OCO-2 soundings have an ACOS retrieved total AOD lower than 0.045.

As we only use OCO-2 soundings with all the best possible flag values, the constraint on ACOS retrieved total AOD had to be loosened for target OCO-2 soundings. However, as shown in the following histogram (Figure R2-1) of the ACOS retrieved total AOD for the selected OCO-2 soundings (without the additional soundings for FOCAL intersection), most of these soundings have low AODs, with an average of 0.08, median of 0.05 and a 75% percentile of 0.1.

We have adapted the manuscript to clarify this point (lines 257-258).

[Figure]

**Figure R2-1.** Distribution of ACOS retrieved total AOD in the selected target OCO-2 soundings

I actually wonder why a sophisticated radiative transfer model such as (V)LIDORT is required at all to perform such calculations. Isn't it just Beer-Lambert's law? Maybe, molecular Rayleigh scattering is included?

Historically, 4A/OP was used to perform radiative transfer in the thermal infrared, and did not take into account polarization. After the extension to NIR and SWIR, 4A/OP now depends on its coupling to VLIDORT to take into account polarization in radiative transfer simulations.

Neglecting polarization in the O2-A band leads to large negative differences in the retrieved surface pressure, compared to the prior surface pressure. VLIDORT is thus used to process this band in order to include Rayleigh scattering for O2 A-band forward simulations, while taking into account polarization effects.

As we do not consider the impact of scattering particles in forward simulations, we can spare the use of (V)LIDORT in the CO2 weak and strong bands as Rayleigh scattering and its polarization effects decrease with the wavelength. This approximation helps to significantly speed up the computations.

These details are included lines 219 – 228 of the revised manuscript.

For example, the differences in spectral residuals between 5AI and ACOS (illustrated in Fig. 10) most likely stem to a large part from the differences in particle treatment, which is corroborated by the finding that 5AI and ACOS retrievals become more similar if the spectral differences are synthetically added to the 5AI processing (section 4.4). Essentially, the ACOS particle treatment is "added" to the 5AI calculations.

Yes, averaged spectral residual account all forward modelling differences between ACOS and 5AI, from scattering particles treatment to spectroscopic parameters, etc.

Given that the remote sensing community has been working on the simultaneous retrievals of greenhouse gases and particle properties for many years, the study lags behind current developments. On the other hand, the paper appears to be one of the first presenting a new algorithm to be applied to the problem and, it is always scientifically interesting and important to compare to new approaches.

If to be published, the authors should clearly discuss the drawback of neglecting particle scattering and they should underline that this assumption is most likely the leading error term. In particular, they should mention it in prominent places such as abstract and conclusions and they should add a section where they describe the (lack of) scattering treatment. Currently, the manuscript reads like (V)LIDORT (with standard scattering treatment) is used and only quite late it becomes clear that particle scattering is neglected.

These comments regarding the discussion of the no-scattering particle hypothesis are in line with those of referee #1. We really appreciate both reviewers stressing the importance of this discussion that was missing in the first submitted manuscript.

As 5AI does enable to take into account the impact of scattering particles in XCO2 retrievals, we took the opportunity of this necessary discussion to perform XCO2 retrievals while taking into account scattering particles, and thus try to assess the forward and inverse consequences of our initial hypothesis. As XCO2

retrievals take longer when considering the impact of scattering particles, especially for OCO-2 that requires using the coupling with VLIDORT, we only processed a few hundreds OCO-2 measurements of our target sounding selection. Because we are interested here in the impact of scattering particles, we focused on 15 target sessions that have collocated TCCON, OCO-2 and also available AERONET data. The independent AERONET information regarding scattering particle optical depth can thus help to discuss the retrieved total aerosol optical depth.

For these retrievals, we took into account two layers of aerosols: a coarse mode layer and a fine mode layer for which we added their respective optical depths in the 5AI state vector. We compared these new 5AI retrieval results to those obtained without considering scattering particles and identified several impacts:

1- reduction or even removal of the 5AI surface pressure airmass dependence, that can be explained by forward and inverse modeling arguments (see Fig. 2, Fig. 8)
2- shift in the averaged 5AI retrieved surface pressure, compared to the prior (see Fig. 8). This partly translates into an averaged XCO2 difference to ACOS that is reduced when taking into account scattering particles (see Fig. 9).

Both 5AI and ACOS retrieved optical depths show a large scatters compared to AERONET data (0.07 and 0.05, respectively).

Regarding the revised manuscript, we separated the Results (Sect. 4) and Discussion (Sect. 5) sections. Subsection 5.1 gives all the details regarding the discussion of the impacts of neglecting scattering particles in 5AI XCO2 retrievals. Of course, the inverse setup choices made for the sub-sample of 5AI retrievals that consider the impact of scattering particles are not exactly identical to ACOS. Differences remain and result in remaining systematic average differences between 5AI and AOCS. We thus kept the averaged – calculated spectral residual discussion, detailed in subsection 5.2, that enables to show that those systematic differences can be compensated.

I understand that full-physics retrievals are tough to get to work, but the authors might want to consider to support their evaluation by some refined assessments wrt. particle scattering:

- For the analysis in section 4, it might be insightful to examine correlation plots of the 5AI-ACOS (and/or 5AI-TCCON) differences with scattering and radiative transfer parameters (e.g. aerosol optical thickness retrieved by ACOS, surface albedo).

The additional retrievals that we performed include some aerosol parameters in the state vector. The impact of this evolution on how 5AI XCO2 results compare to ACOS is thus now directly assessed without relying on correlations.

- The authors could add a forward modelling sensitivity study where they assume a particle scattering scenario (e.g. the one suggested by the ACOS retrievals) for which they actually perform full-physics forward calculations (without the derivatives) and then compare the spectral residuals to clear-sky calculations. Such an assessment could serve as a means to single out the effect of neglecting particle scattering in a clean way.

The additional retrievals that we performed include some aerosol parameters in the state vector. This evolution includes both forward and inverse modelling sensitivity to the inclusion of scattering particle parameters in the state vector.

- ACOS (and other algorithms) previously reported on systematic challenges. For example, there was a bias in earlier ACOS retrievals towards the Southern higher- latitudes which was attributed to stratospheric aerosols (O'Dell et al., 2018). There was a land-ocean bias in early RemoTeC/GOSAT retrievals (Basu et al., 2013). For a future refined assessment, it would be interesting to evaluate 5AI in terms of these specific findings. [O'Dell, C. W, https://doi.org/10.5194/amt-11-6539-2018, 2018, Basu et al., https://doi.org/10.5194/acp-13-8695-2013, 2013.]

Thank you for the suggestion that we will perform in a future evaluation.

Other comments

General: The manuscript repeatedly emphasizes that the 5AI retrieval is a "Bayesian" concept. Is there more to it than just using the Bayesian/optimal estimation formalism (as most regularization concepts do)?

You guessed correctly, we did not mean anything more than the Bayesian / OE formalism. We have adapted the wording to reflect this comment.

A truly Bayesian retrieval would require a careful setup of the prior covariance (to represent the true covariance). Table 1 suggests that the prior covariance is rather a reasonable ad-hoc choice than the true atmospheric co- variance. For example, 4 hPa surface pressure uncertainty is certainly a tight constraint compared to the actual atmospheric variability. Choosing prior covariances ad-hoc is common practice and thus, I do not criticize the procedure per se, but I do recommend being humble when highlighting the Bayesian and "optimal" nature of retrievals under such assumptions.

We do agree on the ad-hoc nature of those choices. Regarding the surface pressure example, it is our understanding that 4 hPa is actually high with regard to the expected uncertainties of ECMWF surface pressure for instance. ACOS ATBD cites [Salstein et al, 2008: https://doi.org/10.1029/2007JD009531] to report 1 – 2 hPa of expected errors. Still this rather high value is an ad-hoc choice. We removed all 'Bayesian' mentions in the manuscript

Introduction: The introductory discussion of other algorithms includes too much of detail. Many of the details (e.g. spectroscopic data, covariance setups, radiative transfer speed-ups) evolve over time without being published. Some key conceptual aspects are missing e.g. the fact that RemoTeC has chosen to not retrieve surface pressure because the problem becomes too ill-posed with respect to the microphysical particle parameters. I would recommend summarizing some of the algorithm features to avoid the paper being already outdated when it gets published. Some important early conceptual work by Oshchepkov et al., 2008, could be added. [Oshchepkov et al., https://doi.org/10.1029/2008JD010061, 2008]

Our intention was to underline the multiplicity of approaches that could be designed to retrieve XCO2 from infrared spectra, from the choice of inverse method, state vector setups, forward model choices and speed-ups and spectroscopic database.

We have adapted the introduction to reflect this great diversity of method without enumerating all of them.

Equation (1): "F" should be bold face, since it is a vector. The comment applies to many occurrences of "F" that follow.

*Thank you for catching this, we have fixed this (lines 123, 125, 172, 178, 184).*

L180: "Epsilon" should be bold face, since it is a vector.

*Thank you for catching this, we have fixed this (line 166).*

L181: "Considering the probability density function instead of vectors" What does the statement mean? Wouldn't it be more appropriate to highlight the key idea of Bayesian statistics? For example: "Assuming that the true state is a particular draw from a Gaussian statistics and that it can be found through a trade-off to a measurement constraint weighted by the respective uncertainties ..." (or better wording).

*We wanted to stress that the formalism is based on random variables. We have rephrased this part (line 168).*

Equation (3): Notation "$(x_a)$" is a bit misleading. It refers to the fact that the derivatives are evaluated at $x_a$ (for the first iteration, if $x_a$ is also the initial guess) while the following parentheses "$(x-x_a)$" refers to a multiplication. I would recommend using a vertical line with subscript $x_a$ to indicate the linearization point of the derivatives.

*We have adapted the notations accordingly (line 178).*

L198: The Levenberg-Marquardt parameter "gamma" is required to vanish at the final iteration since otherwise it will induce a (probably undesired) regularization of the problem. In my experience, not all readers are aware of this effect and thus, I would recommend mentioning it clearly.

*Indeed, we did not mention how gamma evolves along iterations. We implemented the evolution based on a ratio of cost functions described in ACOS ATBD (Version 3.0, Rev 0, pages 53-54). In converging retrievals, gamma thus decreases nicely towards 0. We have included this comment in the revised manuscript (lines 186-187).*

L199: It would be interesting to quantify the advantage in computational cost when not updating the derivatives. Probably, not updating the derivatives means that convergence is somewhat slower and thus, more iterations are required. So, there is a trade-off between number of iterations and cost per iteration. Could you give rough numbers?

*The computational time critical step for a 5AI OCO-2 XCO2 retrieval is currently the computation of the Jacobian matrix when we use 4A/OP coupling with VLIDORT. Once the Jacobian matrix calculation is complete, forward computations without Jacobian matrix update are practically cost-less with regard to this critical step, so the trade-off is very easily settled compared to calculating several time this matrix, even if a few additional iterations are required.*

*When updating the Jacobian matrix, it takes about 8 times longer to reach convergence for one sounding, just because of the several VLIDORT Jacobian calculations in the O2 A-band.*

Equation (6): A downside of not updating the Jacobians is that the averaging kernels are calculated with respect to the prior state (not the iterated retrieved state). The prior state is typically "further away" from the true state than the retrieval and thus, it is in the non-linear regime. I think this is a minor issue in general, but it might cause confusion when, for example, evaluating the effects of non-linearity on retrievals. One might consider calculating the Jacobians for the first and the last iteration to get rid of

this effect.

This is indeed a minor issue. However, the overall shape of the column averaging kernel is not expected to evolve strongly along iterations, especially as neglecting the impact of scattering particles makes the problem way more linear. When taking into account scattering particles, the Jacobian matrix is computed for all iterations, thus resolving the issue.

L263: For a retrieval method paper, comparisons to raw retrievals of other algorithms are the essential tool. Comparisons to bias-corrected retrievals will not inform on methodological issues, but, to a large extent, such comparisons will mirror the effects of the bias corrections, in particular since some algorithms require substantial bias corrections. So, the (anyway sparse) comparisons to FOCAL could be removed from the manuscript.

We have followed your advice and removed the discussion with FOCAL data. The revised manuscript has been adapted accordingly.

Fig. 10. I recommend plotting the residuals and transmittances in the same units (either absolute radiance or relative transmittance) to allow for comparison of the amplitude of the residuals wrt. the spectra.

We have adapted the figure with spectral residuals plotted in transmission (line 449).